# Initiating polyketide biosynthesis by on-line methyl esterification

Pengwei Li[1], Meng Chen[1,2], Wei Tang[1,2], Zhengyan Guo[1], Yuwei Zhang[1,2], Min Wang[1,2,3], Geoff P. Horsman[4], Jin Zhong[1], Zhaoxin Lu[5] & Yihua Chen[1,2✉]

Aurantinins (ARTs) are antibacterial polyketides featuring a unique 6/7/8/5-fused tetracyclic ring system and a triene side chain with a carboxyl terminus. Here we identify the *art* gene cluster and dissect ART's *C*-methyl incorporation patterns to study its biosynthesis. During this process, an apparently redundant methyltransferase Art28 was characterized as a malonyl-acyl carrier protein *O*-methyltransferase, which represents an unusual on-line methyl esterification initiation strategy for polyketide biosynthesis. The methyl ester bond introduced by Art28 is kept until the last step of ART biosynthesis, in which it is hydrolyzed by Art9 to convert inactive ART 9B to active ART B. The cryptic reactions catalyzed by Art28 and Art9 represent a protecting group biosynthetic logic to render the ART carboxyl terminus inert to unwanted side reactions and to protect producing organisms from toxic ART intermediates. Further analyses revealed a wide distribution of this initiation strategy for polyketide biosynthesis in various bacteria.

[1] State Key Laboratory of Microbial Resources & CAS Key Laboratory of Microbial Physiological and Metabolic Engineering, Institute of Microbiology, Chinese Academy of Sciences, Beijing, China. [2] University of Chinese Academy of Sciences, Beijing, China. [3] School of Biotechnology and Health Sciences, Wuyi University, Jiangmen, Guangdong, China. [4] Department of Chemistry and Biochemistry, Wilfrid Laurier University, Waterloo, ON, Canada. [5] College of Food Science and Technology, Nanjing Agriculture University, Nanjing, China. ✉email: chenyihua@im.ac.cn

Polyketides are a large family of structurally diverse natural products with varied biological and pharmacological activities including antibacterial, antitumor, and immunosuppressant[1,2]. A great number of polyketides are constructed by giant modular polyketides synthases (PKSs) with multifunctional domains. After initiated by the loading module, the polyketide chain will be elongated iteratively by each extension module in a manner analogous to fatty acid biosynthesis. The acyltransferase (AT) domain loads a specific extender unit to acyl carrier protein (ACP) and the ketosynthase (KS) domain performs a two-carbon addition to ACP-tethered acyl-thioesters via head-to-tail Claisen condensation. The newly synthesized $\beta$-ketothioester intermediates could be modified by additional domains (e.g., ketoreductase (KR), dehydratase (DH), enoyl reductase (ER), and methyltransferase (MT)) in the extension module before they are released by an offloading module which usually contains a thioesterase (TE) or a reductase (R) domain[3–5]. Specifically, a large family of modular PKSs is *trans*-AT PKS, which does not have an AT domain in each extension module as *cis*-AT PKS but shares a standalone AT by multiple extension modules[6–8]. In principle, the polyketide core structures are consistent with the domain organizations of module PKSs, which was called the co-linearity rule[9]. Notably, *trans*-AT PKS frequently deviates from this rule owing to the presence of modules with aberrant domain architecture or no apparent function. The phylogeny and substrate specificity of their KS domains can be applied for the prediction of product structures of *trans*-AT PKS[10,11]. The vast structural diversity of polyketides is introduced by different combinations of initiation, extension, and termination steps as well as versatile tailoring processes. For the initiation process, malonyl-coenzyme A (CoA) is a frequently used starter unit, which is usually decarboxylated to an acetyl-thioester to begin the extension process. Consequently, most polyketides constructed in this way have a terminal methyl group unless it is modified by tailoring enzymes[12–14].

Aurantinins (ARTs) are a group of four antibiotics isolated from *Bacillus aurantinus* strain KM-214 and *Bacillus subtilis* fmb60[15–18] all sharing a unique 6/7/8/5-fused tetracyclic ring system and a triene side chain with an unusual carboxyl terminus, but differing in glycosylation and methylation patterns (Fig. 1a). Of the four known compounds, ART B is the major product and possesses a special 3′-keto-$\beta$-sugar attached to its C-17 hydroxyl group. ARTs exhibit broad activity against Gram-positive bacteria including multidrug-resistant *Staphylococcus aureus* and anaerobic bacteria like *Clostridium sporogenes*, and they kill bacteria by disrupting the cell wall membrane[18]. Biosynthetic insight from previous feeding experiments with isotope-labeled precursors suggested that the ART B aglycone is assembled from two unusual polyketide chains: (i) a short chain containing an uncommon tail-to-tail condensation of two acetate units (C-24 to C-27); (ii) a long dicarboxylic acid chain with one of the two carboxyl groups (C-1 carboxyl of ART) derived from the C-1 of an acetate unit (Fig. 1a)[17]. However, without knowledge of the ART biosynthetic gene cluster it remains unclear how the two ART polyketide chains are assembled and undergo the unusual cyclization and glycosylation processes.

In this study, we identified the *art* gene cluster in *B. subtilis* fmb60 and proposed the ART biosynthetic pathway by dissecting *C*-methyl incorporation patterns. We were particularly intrigued by the presence of a methyltransferase (MT) gene *art28* that is indispensable for ART biosynthesis but apparently 'redundant' with respect to the known *C*-methyl incorporations; i.e., the other methyltransferases could account for all SAM-dependent methylations on the ART scaffold. Characterization of Art28 as a malonyl-acyl carrier protein (ACP) *O*-MT revealed an alternative initiation strategy for polyketide biosynthesis involving malonyl group transfer to ACP$_{Art10}$ followed by on-line methyl esterification to trigger polyketide chain extension. This methyl ester 'protecting group' is retained throughout ART biosynthesis and is hydrolyzed by a membrane protein, Art9, to expose the free carboxyl terminus at the final step. The methyl ester bond formation and cleavage process catalyzed by Art28 and Art9 are reminiscent of carboxyl esterification and deprotection strategies frequently used in chemical synthesis. Moreover, the inactive biosynthetic intermediate ART 9B is converted to the active antibiotic ART B via Art9-catalyzed hydrolysis of the methyl ester, indicating that the terminal methyl ester can also protect the ART producers from toxic intermediates. In silico and in vitro analyses of the Art28-like *O*-MTs revealed that this on-line methyl esterification initiation strategy for polyketide biosynthesis is distributed in different bacteria from four phyla.

## Results

**Identification of the ART biosynthetic gene cluster**. Two polyketide biosynthetic gene clusters were found in the genome of *B. subtilis* fmb60 (accession: NZ_LYMC01000002)[18]. A *trans*-AT polyketide synthase (PKS) gene cluster[6,7] was proposed as the ART biosynthetic gene cluster (Fig. 1b) by in silico analyses of the PKS domains, and was validated by the fact that disruption of *art11*, a key PKS gene, totally abolished ART production (Fig. 1c). The boundaries of the *art* gene cluster were then determined by sequential inactivation of the flanking genes. The right boundary of the *art* gene cluster was assigned between the genes *art28* and *orf(+1)* because ART production was completely blocked by in-frame deletion of the former gene but was not obviously influenced by knocking-out the latter. The left boundary of the *art* gene cluster was assigned between *art1* and *orf(−1)* similarly according to their gene inactivation results. The transcription anti-terminator Art1 was proposed to be a positive regulator based on the fact that no ART production was detected in the *B. subtilis* Δ*art1* mutant (Fig. 1c). Thus, the *art* gene cluster was narrowed down to an 80.3-kb region of DNA containing 28 genes, with 17 of them PKS-related (Supplementary Table 1).

**Investigation of the *C*-methyl incorporation patterns of ARTs**. In silico analysis results of the *art* gene cluster using bioinformatic tools like TransATor[10] were informative but provided few clues regarding the two-polyketide chain assembly model and the unusual starter units of the polyketide chains. We therefore sought to begin elucidating the biosynthetic pathway by first dissecting *C*-methyl incorporation patterns of ART. The ART B aglycone contains seven *C*-methyl groups, five of which are derived from *S*-adenosyl-L-methionine (SAM) and two from the C-2 of acetate[17] (Fig. 1a). Our analysis of the *art* gene cluster revealed ten genes putatively involved in methyl incorporations including two standalone MT genes (*art4* and *art28*), four genes (*art11*, *art13*, *art14*, and *art17*) encoding PKSs with a MT domain, and five genes (*art7*, *art18*, *art19*, *art20*, and *art21*) encoding proteins of the $\beta$-branching system (Supplementary Table 1). MTs generally use SAM as a precursor[19]; in contrast, the $\beta$-branching system appends the C-2 of an acetate unit to the $\beta$-position of a polyketide chain to form a *C*-methyl appendage by sequential condensation, decarboxylation, and dehydration reactions[20,21].

Each of the four PKS-MT domains was inactivated by a point mutation at its conserved active site His residue (His → Ala). All four mutants lost the capacity to produce ART A-D, and three of them, except *B. subtilis* Δ*art17MT*, accumulated ART congeners (Fig. 1c) with MS and NMR profiles consistent with demethyl-ART A or B (Supplementary Figs. 1–4 and Supplementary Tables 2–4). Specifically, *B. subtilis* Δ*art11MT* accumulated ART

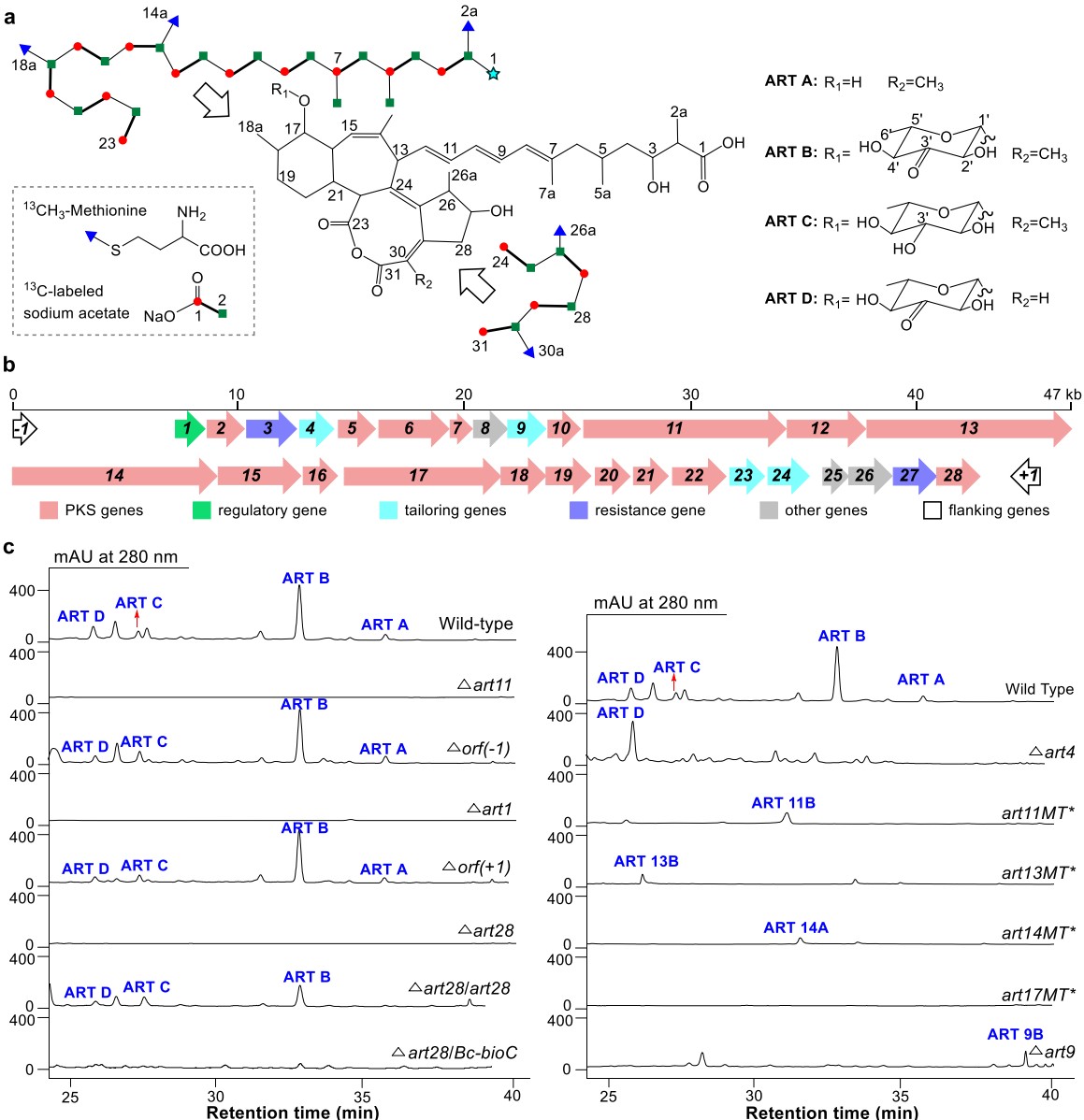

**Fig. 1 Structures and the biosynthetic gene cluster of ARTs. a** Structures of ART A–D. The isotope labeling pattern of the ART B aglycone is shown as two separate polyketide chains. The C-1 carbon (marked with a light blue star) was previously proposed to originate from the C-1 of acetate[17]; in this study, we propose it originates from the free carboxyl end of the malonyl starter unit. **b** Genetic organization of the *art* gene cluster. **c** HPLC profiles of *B. subtilis* fmb60 wild-type and mutant strains. *B. subtilis* Δ*art28/art28* and Δ*art28/Bc-bioC* are two complementation strains of the *art28* in-frame deletion mutant *B. subtilis* Δ*art28*.

11B lacking the C-2 methyl; *B. subtilis* Δ*art13MT* accumulated ART 13B without the C-14 methyl; *B. subtilis* Δ*art14MT* accumulated aglycone ART 14A devoid of the C-18 methyl, indicating that C-18 methylation is required for glycosylation (Fig. 2). Productions of all the demethylated ART congeners were reduced dramatically in those mutant strains with PKS mutagenesis. These results allowed us to propose that PKSs Art10-15 assembles the long polyketide chain following the co-linearity rule[9] (Fig. 2). This was also supported by the analyses of the five β-branching genes *art7*, *art18*, *art19*, *art20*, and *art21*. Loss of ART production when any one of four genes (*art18-art21*) was in-frame deleted indicated that the β-branching system is necessary for ART biosynthesis. In silico analyses revealed that all three ACP domains of modules 3 and 4 in Art11 possess the conserved Trp residues diagnostic of the 3-hydroxy-3-methylglu-taryl-CoA synthase of the β-branching system[22] (Supplementary

Fig. 5), implying that the methyl groups at C-5 and C-7 are incorporated into the intermediates generated on Art11 by the β-branching enzymes together.

The short ART polyketide chain was hypothesized to be assembled by the remnant PKS Art17 (Fig. 2). The *art* gene cluster contains two acyltransferase genes *art2* and *art6*. A phylogenetic analysis indicated that Art6 is a typical malonyl-ACP acyltransferase of *trans*-AT PKS; while Art2 fell into the same clade with EtnB and GdnB (Supplementary Fig. 6), two putative succinyl-ACP transferases involved in the biosynthesis of etnangien[23] and gladiolin[24], respectively. We hypothesized that Art2 initiates the short chain biosynthesis by loading an uncommon starter unit succinyl to the first ACP of Art17. After a methylation by the MT domain of Art17 (adding the C-26 methyl of ARTs), it is elongated twice by modules I and II to afford the short polyketide chain.

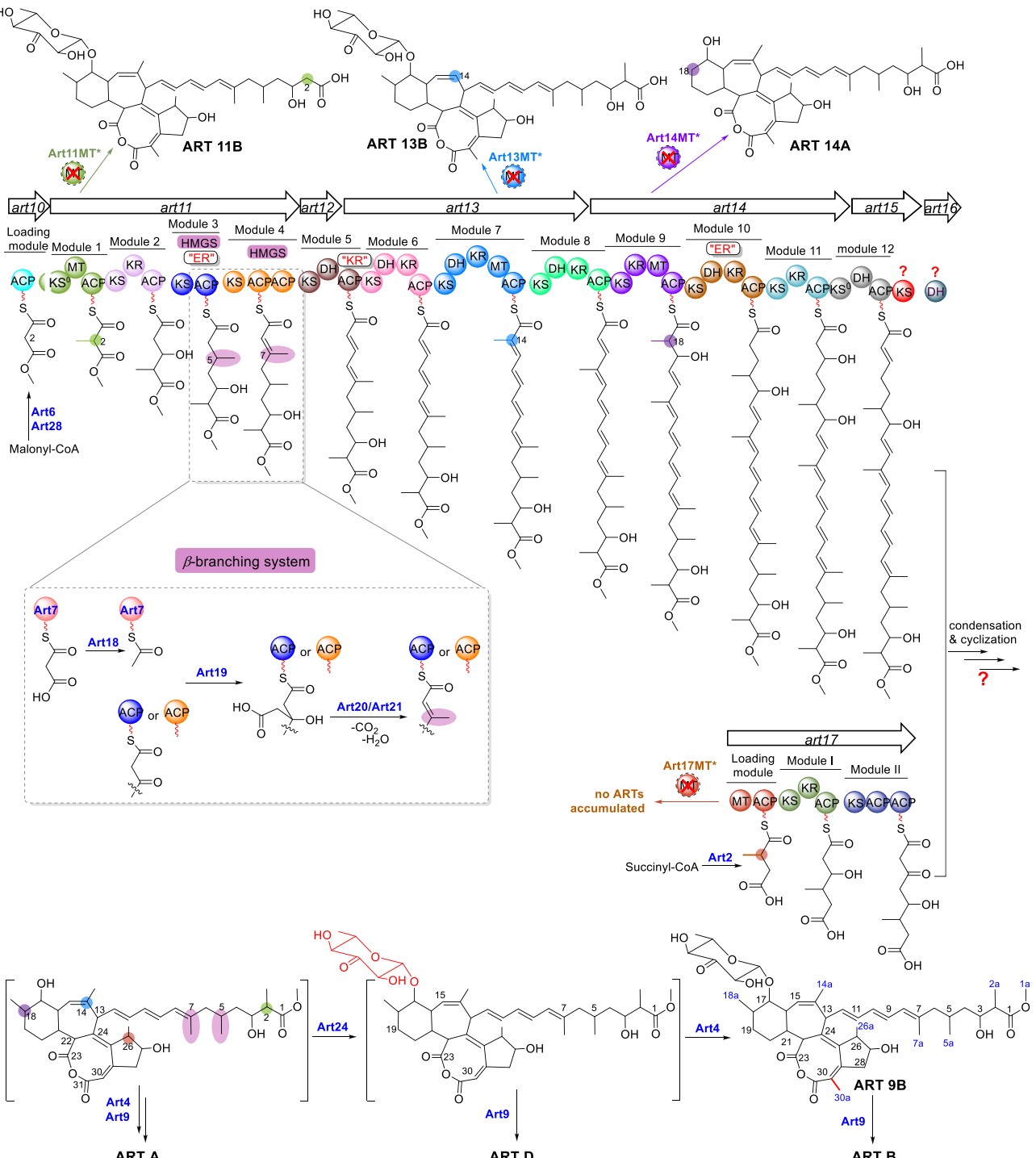

**Fig. 2 The proposed biosynthetic pathway of ARTs based on the two-polyketide-chain assembly model.** The inset depicts the process that methyl groups are appended to C-5 and C-7 of the ACP-bound intermediates by *β*-branching system. The methyl group at C-5 is first installed on the *β*-carbonyl of nascent polyketide chain tethered on ACP (dark blue) in module 3, subsequently, C-7 methylation was formed on the intermediate tethered on module 4 ACP (orange) in a same manner. The absence of methyl groups at C-2, C-14, and C-18 were highlighted with points in green, blue, and purple corresponding to the inactivated domains Art11MT, Art13MT, and Art14MT, respectively. ACP acyl carrier protein, KS ketosynthase, KS⁰ non-elongating ketosynthase, KR ketoreductase, DH dehydratase, ER enoylreductase, MT methyltransferase, HMGS hydroxymethylglutaryl synthase.

The two standalone MT genes were then studied to inspect the C-30 methylation mechanism. Although Art4 and Art28 share only 15.2% identity, they both display significant similarities with proteins annotated as UbiE, a large family of MTs governing the *C*-methylation of ubiquinone or menaquinone[25]. In-frame deletion of *art4* abolished production of ART A and B but

yielded more ART D (Fig. 1c, Supplementary Fig. 7, and Supplementary Table 5), implying that Art4 catalyzes C-30 methylation. Overall, the incorporation of all seven *C*-methyl groups of ART B could be reasonably assigned leaving Art28 as a 'redundant' MT. However, this was inconsistent with our finding during gene cluster boundary assignment that the *art28* gene is

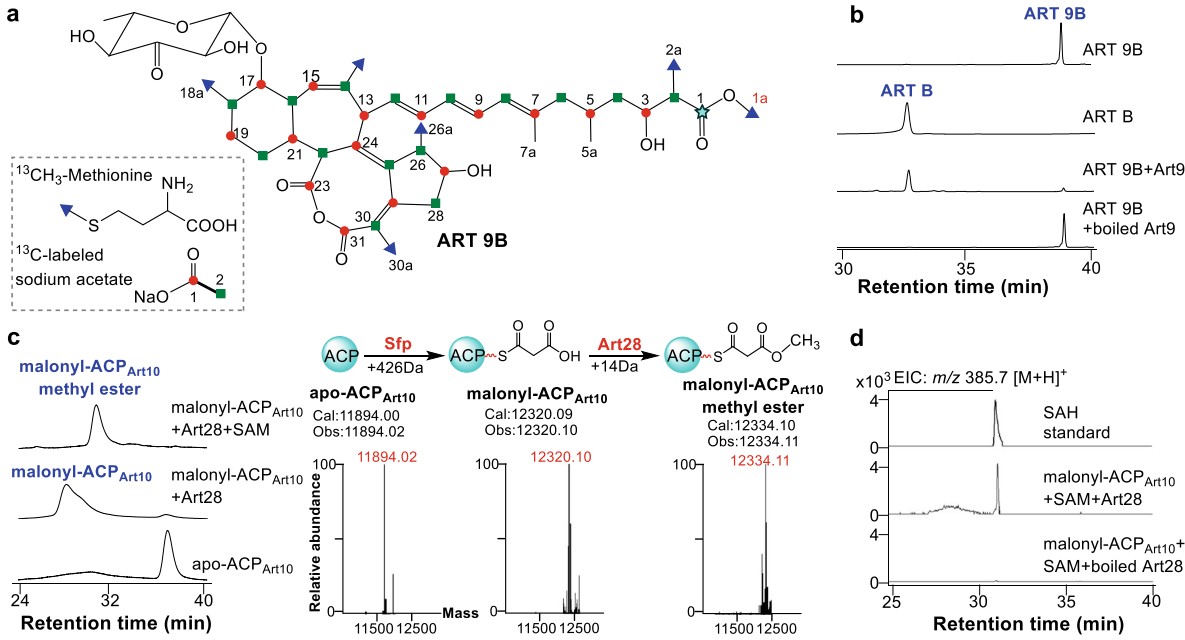

**Fig. 3 Characterization of Art9 as an esterase and Art28 as an *O*-MT. a** The isotope labeling pattern of ART 9B with different [13]C-labeled precursors. The C-1 carbon (marked with a light blue star) is proposed to originate from the free carboxyl end of the malonyl starter unit. **b** HPLC analysis of the enzymatic assays of Art9. **c** HPLC detection of the methyl esterification of malonyl-ACP[Art10] in enzymatic assays of Art28. Deconvoluted mass spectra of apo-ACP[Art10], malonyl-ACP[Art10], and malonyl-ACP[Art10] methyl ester are shown. **d** LC-MS analysis of SAH production in the enzymatic assays of Art28. EIC extracted ion count.

necessary for ART biosynthesis. Unfortunately, the absence of any accumulated ART congener from *B. subtilis* Δ*art28* did not provide insight regarding the function of Art28.

**Characterization of Art9 as an esterase exposing the terminal carboxyl group.** Previous feeding experiments showed that that the terminal free carboxylic acid of ART is derived from the C-1 of a cleaved acetate unit[17], suggesting a possible origin for the free terminal carboxylic acid group by oxidative C–C bond cleavage of the polyketide chain and subsequent hydrolysis. Such oxidative cleavages have been observed in oocydin A and viriditoxin produced by *Serratia plymuthica* 4Rx13 and *Paecilomyces variotii*, respectively, where the Baeyer-Villiger monooxygenases (BVMO) OocK and VdtE inserted an oxygen into the polyketide chain between the two carbons of one acetate unit[26,27]. The ester bonds can be further hydrolyzed by putative esterases to expose the terminal carboxyl group. Interestingly, although no oxygenase gene could be found in the *art* gene cluster, there is an esterase gene *art9* that encodes a protein sharing 33% identity with BioH, a methyl esterase exposing the ω-carboxyl group of pimeloyl-ACP during biotin biosynthesis[28]. The *art9* gene in-frame deletion mutant *B. subtilis* Δ*art9* was incapable of synthesizing ART A-D but produced ART 9B (Figs. 1c and 3a), the C-1 methyl ester derivative of ART B (Supplementary Fig. 8 and Supplementary Table 6), which implied that Art9 is the esterase hydrolyzing ART 9B to afford ART B. The *N*-His[6]-Art9 was then expressed in *Escherichia coli*, purified by affinity chromatography (Supplementary Fig. 9a), and incubated with ART 9B. It could hydrolyze ART 9B to ART B efficiently, verifying that Art9 is the esterase liberating the terminal carboxyl groups of ARTs (Fig. 3b).

If the terminal methyl ester structure of ART 9B is formed by the insertion of an oxygen as occurs in oocydin A and viriditoxin biosynthesis, the terminal *O*-methyl group (1a position of ART 9B) should be derived from the C-2 of an acetate unit. However, when we fed *B. subtilis* Δ*art9* with [2-13C] acetate, the isolated

ART 9B shared the same labeling pattern as ART B, but its terminal *O*-methyl was not [13]C labeled as predicted, which is not in agreement with the BVMO-catalyzed oxygen insertion mechanism. Further isotope labeling feeding experiments showed that the [13]C NMR signal of C-1 was not obviously enriched by [1-13C] acetate, while the C-1a signal of the terminal *O*-methyl was clearly enriched by [13CH3]-methionine (Fig. 3a, Supplementary Fig. 10 and Supplementary Table 7). To confirm this, we carried out [1-13C] acetate feeding experiment of *B. subtilis* fmb60 and observed that ART B was [13]C labeled at positions as reported previously[17], except that C-1 was not clearly enriched by [1-13C] acetate (Supplementary Fig. 11). Overall, these results suggested that the terminal methyl ester of ART 9B is not derived from inserting an oxygen in an acetate unit but is more likely established by a MT using SAM as a substrate, which prompted us to revisit the role of the 'redundant' MT Art28 in the *art* gene cluster.

**Characterization of Art28 as a malonyl-ACP *O*-MT.** Careful reanalysis of Art28 revealed that, besides UbiE proteins, it is also similar to proteins annotated as BioC such as the well-characterized *Bc*-BioC (19.2% identity) from *Bacillus cereus* ATCC10987[29], which acts as an *O*-MT that catalyzes the methyl esterification of malonyl-ACP to initiate biotin biosynthesis. To probe the function of *art28*, *B. subtilis* Δ*art28* was complemented by expressing *art28* or the *Bc-bioC* gene *in trans*. Although *Bc-bioC* was not as efficient as *art28*, both genes restored production of ARTs (Fig. 1c, Supplementary Fig. 12), suggesting that Art28 has the same function as *Bc*-BioC.

We therefore proposed Art28 as a malonyl-ACP *O*-MT initiating chain extension of the long polyketide by forming a malonyl-ACP methyl ester. The substrate of Art28 could be the malonyl-ACP of protein Art10, which has an *N*-terminal ACP domain. Both Art28 and the truncated apo-ACP[Art10] (residues 1–87) were expressed as *N*-His[6] tagged proteins in *E. coli* and

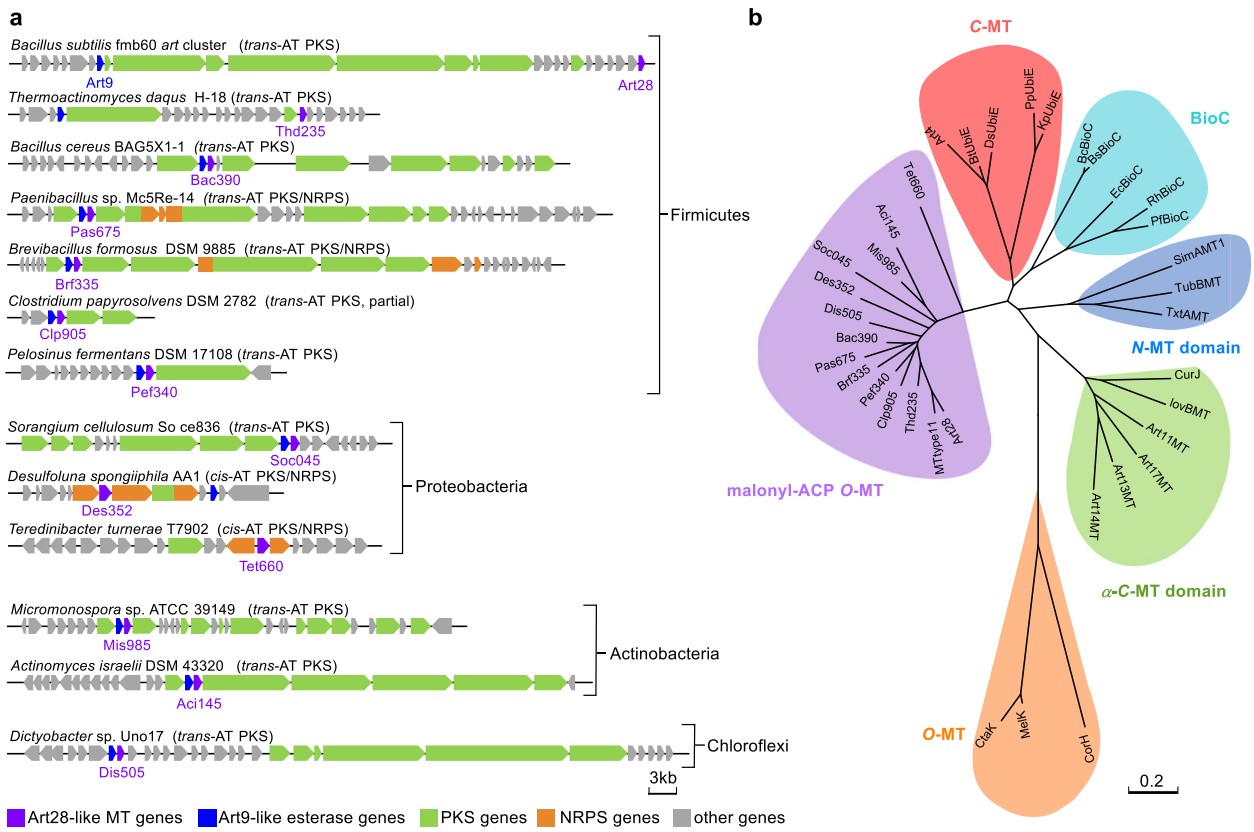

**Fig. 4 Distribution and phylogenetic analysis of the Art28 homologs. a** Schematic representatives of the biosynthetic gene clusters containing genes encoding Art28 homolog proteins. The Art28-like MT genes are shown in purple and the Art9-like esterase genes are shown in blue. **b** Phylogenetic analysis of different types of MTs (the details of related proteins are listed in Supplementary Table 8).

purified by affinity chromatography (Supplementary Fig. 9a). A phosphopantetheinyl transferase Sfp[30] with high substrate promiscuity was used to convert apo-ACP$_{Art10}$ to malonyl-ACP$_{Art10}$ using malonyl-CoA as a substrate. When Art28 was incubated with malonyl-ACP$_{Art10}$ and SAM, efficient formation of malonyl-ACP$_{Art10}$ methyl ester and $S$-adenosyl-L-homocysteine (SAH) could be detected by HPLC and LC-MS, respectively (Fig. 3c, d). If the assay was performed without SAM, no malonyl-ACP$_{Art10}$ methyl ester was formed. To test whether malonyl-CoA can be recognized by Art28, it was used to replace malonyl-ACP$_{Art10}$ in the Art28 assay, and no malonyl-CoA methyl ester and SAH were detected by LC-MS (Supplementary Fig. 13). These results verified that Art28 is a malonyl-ACP $O$-MT. Taken together, Art28 and Art9 form an enzyme pair that catalyzed two 'cryptic' reactions to shape the rare terminal carboxyl group of ARTs. Moreover, Art28 represents an unusual initiation strategy of polyketide biosynthesis, in which polyketide chain extension is triggered by the on-line methyl esterification of malonyl-ACP.

**Distribution of the Art28 homologs**. To find out that whether the on-line methyl esterification initiation strategy is adopted by other PKSs, we performed a ClusterBlast against the antiSMASH database[31] using Art28 as a probe, which yielded 12 biosynthetic gene clusters (BGCs) containing genes encoding Art28 homolog proteins (Fig. 4a). The *art* gene cluster and the 12 resultant BGCs are widely distributed in bacteria including Firmicutes (7), Proteobacteria (3), Actinobacteria (2), and Chloroflexi (1). Among the 13 BGCs, 9 are *trans*-AT PKS gene clusters, 2 are hybrid *cis*-AT PKS/NRPS (non-ribosomal peptide synthetase) gene clusters, and the other 2 are hybrid *trans*-AT PKS/NRPS gene clusters. Significantly, the genes encoding Art9 homologs were observed in

11 of the 12 BGCs, indicating that the methyl ester bond is hydrolyzed to expose the carboxyl terminus in most cases. The hybrid PKS/NRPS cluster in *Teredinibacter turnerae* T7902 may generate products with a methyl ester terminus like ART 9B, since no *art9* ortholog was observed in this BGC.

A phylogenetic analysis revealed that Art28 and its 12 homologs were clustered in one clade clearly separated from the other MTs (Fig. 4b and Supplementary Table 8), indicating they have similar and distinctive functions. To test the malonyl-ACP $O$-MT activity of the Art28 homologs, 5 of them (Thd235 from *Thermoactinomyces daqus* H-18, Bac390 from *Bacillus cereus* BAG5X1-1, Pas675 from *Paenibacillus* sp. Mc5Re-14, Brf335 from *Brevibacillus formosus* DSM 9885, and Clp905 from *Clostridium papyrosolvens* DSM 2782) were overexpressed in *E. coli*, purified as $N$-His$_6$-tagged proteins (Supplementary Fig. 9b), and assayed along with Art28. All five proteins could efficiently convert malonyl-ACP$_{Art10}$ to its methyl ester (Supplementary Fig. 14), implying that they play the same role as Art28 in natural product biosynthesis. Overall, the widespread occurrence of Art28 homologs suggests that malonyl-ACP methyl esterification is a polyketide biosynthesis initiation strategy adopted by various bacteria.

**Antibacterial activities of the ART congeners**. Finally, the antibiotic activities of the ART congeners were tested against different Gram-positive and Gram-negative bacteria (Table 1). The obtained data clearly showed that ARTs only inhibit growth of Gram-positive bacteria and the antibacterial activities of demethyl-ART congeners are reduced. Intriguingly, ART 9B did not show any antibacterial activity revealing that exposing the terminal carboxyl group is critical to the antibacterial activities of

**Table 1 Antibacterial activities of ARTs.**

| Samples | MIC values (μM) | | | | | | |
|---|---|---|---|---|---|---|---|
| | BS | BC | SA | MRSA | SM | PA | EC |
| ART B | 3.12 | 3.12 | 1.56 | 1.56 | 3.12 | 100 | >100 |
| ART 11B | 12.5 | 12.5 | 6.25 | 3.12 | 12.5 | >100 | >100 |
| ART 13B | 12.5 | 12.5 | 6.25 | 3.12 | 12.5 | >100 | >100 |
| ART D | 25 | 25 | 12.5 | 6.25 | 12.5 | >100 | >100 |
| ART 9B | >100 | >100 | >100 | >100 | >100 | >100 | >100 |
| ART A | 12.5 | 12.5 | 12.5 | 6.25 | 12.5 | >100 | >100 |
| ART 14A | 50 | 50 | 50 | 50 | 100 | >100 | >100 |
| Erm | 0.78 | 0.78 | 1.56 | >100 | 0.2 | 100 | 100 |

*BS Bacillus subtilis BS168, BC Bacillus cereus CGMCC 1.0230, SA Staphylococcus aureus ATCC 6538, MRSA methicillin-resistant Staphylococcus aureus strain 113, SM Streptococcus mutans UA159, PA Pseudomonas aeruginosa, EC Escherichia coli CGMCC 1.12873, Erm erythromycin.*

ARTs. When the cell membrane permeability of *S. aureus* ATCC 6538 was evaluated using propidium iodide, a fluorescent dye binding nucleic acid, it was showed that ART B could damage the cell membrane in a dose-dependent manner and displayed good activity at a concentration as low as 4 μM. In comparison, ART 9B, with its terminal carboxyl group methyl esterified, had little effect on the cell membrane permeability of *S. aureus* ATCC 6538 even at a concentration as high as 100 μM (Supplementary Fig. 15).

## Discussion

Previous isotope feeding studies suggested a convergent biosynthetic pathway fusing two separate polyketides, and our current results built on this work establish a common ART biosynthetic pathway as depicted in Fig. 2. Biosynthesis of the long polyketide chain is initiated by the standalone malonyl-CoA:ACP acyltransferase Art6, which loads a malonyl unit to the *trans*-AT PKS ACPs including $ACP_{Art10}$ in the loading module. Subsequently, the *O*-MT Art28 triggers polyketide chain extension by catalyzing the methyl esterification of malonyl-$ACP_{Art10}$. Among the 12 PKS modules in Art11-Art15, the ketosynthase (KS) domains of modules 1 (Art11KS1) and 12 (Art14KS4) were proposed to be $KS^0$ domains with transthiolation instead of condensation activity: Art11KS1 lacks approximately 130 residues at its *N*-terminus, and Art14KS4 contains an Asn in place of the first His residue of the conserved Cys-His-His catalytic triad (Supplementary Fig. 16). Substitutions of this His residue have been observed in several $KS^0$ domains such as OzmH-KS8[32] and BaeKS3[8]. We therefore propose that the malonyl methyl ester tethered on $ACP_{Art10}$ is transferred to the ACP of module 1, methylated at C-2, and then elongated ten times by modules 2 to 11 to assemble the long polyketide chain. A final transfer to the ACP of module 12 allows dehydration to form the double bond between C-21 and C-22. We propose that biosynthesis of the short polyketide chain is initiated by Art2-catalyzed loading of a succinyl unit onto the first ACP of Art17. After *C*-methylation and two elongation steps, the short polyketide chain is condensed with the long chain to generate the ART polyketide skeleton, which then undergoes multiple cyclization and modification steps to furnish the unique 6/7/8/5-fused tetracyclic ring system. However, we cannot exclude the possibility that the ART polyketide skeleton originates from a single large polyketide chain generated by Art10-Art17 following the co-linearity rule; this one-polyketide-chain biosynthetic model is shown in Supplementary Fig. 17. Further studies on the ART PKSs, especially on the KS domain of Art15 and the dehydratase Art16, will be conducted to determine which pathway is correct. We propose three subsequent tailoring steps to fashion ART A-D: Art24 performs glycosylation at 17-OH;

Art4 adds a methyl group at C-30; and Art9 hydrolyzes the terminal methyl ester bond to expose the C1 carboxyl terminus.

A striking feature of ART biosynthesis is the on-line methyl esterification initiation strategy, which is unusual for polyketide biosynthesis. Polyketide chain extension generally starts immediately after the starter unit is loaded; the initiation process includes preparation of the acyl-CoA starter unit (if it is not available from primary metabolism) and transacylation of the starter unit to the loading ACP[12,14,33]. However, in some cases the starter unit must be modified after being loaded onto an ACP in order to trigger polyketide chain extension. One classic example is on-line decarboxylation of the malonyl or methylmalonyl starter unit. In *cis*-AT modular PKSs, it is catalyzed by the *N*-terminal $KS^Q$ domain of PKSs[34–37] (Fig. 5a), while in *trans*-AT modular PKSs, it is performed by a GCN5-related *N*-acetyltransferase (GNAT)-like domain[38,39] (Fig. 5b). In addition, an on-line transamidation strategy has been proposed to initiate biosynthesis of glutarimide-containing compounds such as isomigrastatin and cycloheximide, although this mechanism requires further validation by in vitro studies[40,41] (Fig. 5b). Here we show that the malonyl-ACP *O*-MT Art28 represents an alternative polyketide initiation strategy, in which the malonyl starter unit undergoes an on-line methyl esterification prior to chain extension. This initiation logic installs a terminal methyl ester on polyketides, and in the case of the ARTs, the methyl ester is hydrolyzed by Art9 to expose the terminal free carboxylic acid as the last biosynthetic step (Fig. 5b). Discovery of this malonyl-ACP methyl esterification initiation strategy expands the priming repertoire of polyketide biosynthesis and enables the discovery of 12 analogous natural product biosynthetic gene clusters by genome mining. Significantly, 11 of the 12 biosynthetic gene clusters possess an Art9-like esterase, indicating that most compounds generated via this initiation strategy have an unusual carboxyl terminus. We are currently exploring several Art28-Art9-type protection–deprotection systems as synthetic biology tools for expedient elaboration of polyketide carboxyl termini.

The esterification status of the carboxyl termini is critical for biological activity: the methyl ester ART 9B exhibits no inhibitory activity, but the free acid analog ART B can inhibit Gram-positive bacteria by disrupting cell membrane and causing leakage of intracellular components[18]. Introduction of the methyl ester group at the beginning of ART biosynthesis and hydrolyzing it at the end is reminiscent of traditional chemical synthetic protecting group strategies, and may similarly serve to prevent the unusual carboxylic acid from engaging in unwanted side reactions during the biosynthesis. In addition, the methyl ester group may also protect the producing organisms (*Bacillus* is also Gram-positive bacterium) from toxic ART intermediates. As exemplified by ART 9B, which could not disrupt the cell membrane of *S. aureus*, methyl esterification of the ART intermediates may influence their interactions with cell membrane by changing their hydrophilicity statuses dramatically. Such a phenomenon was described previously for the ionophore antibiotic zincophorin, which also has a carboxyl terminus. The chemically synthesized zincophorin analog with a terminal methyl ester modification exhibited no antibacterial activity[42]. Once the ART 'pro-drugs' are activated, they may be pumped out immediately by the efflux proteins, Art3 and Art27. This assumption is supported by the facts that (i) Art9 is a membrane located esterase with a transmembrane region (Supplementary Fig. 18); and (ii) ARTs are almost exclusively distributed in the fermentation broth but not inside cells (Supplementary Fig. 19). Similar protecting group strategy has been observed during the biosynthesis of a number of natural products and versatile activation strategies have been recruited. For example, the 'pro-drug' of antibiotic xenocoumacin is activated via cleavage of an acylated D-asparagine by a membrane-anchored

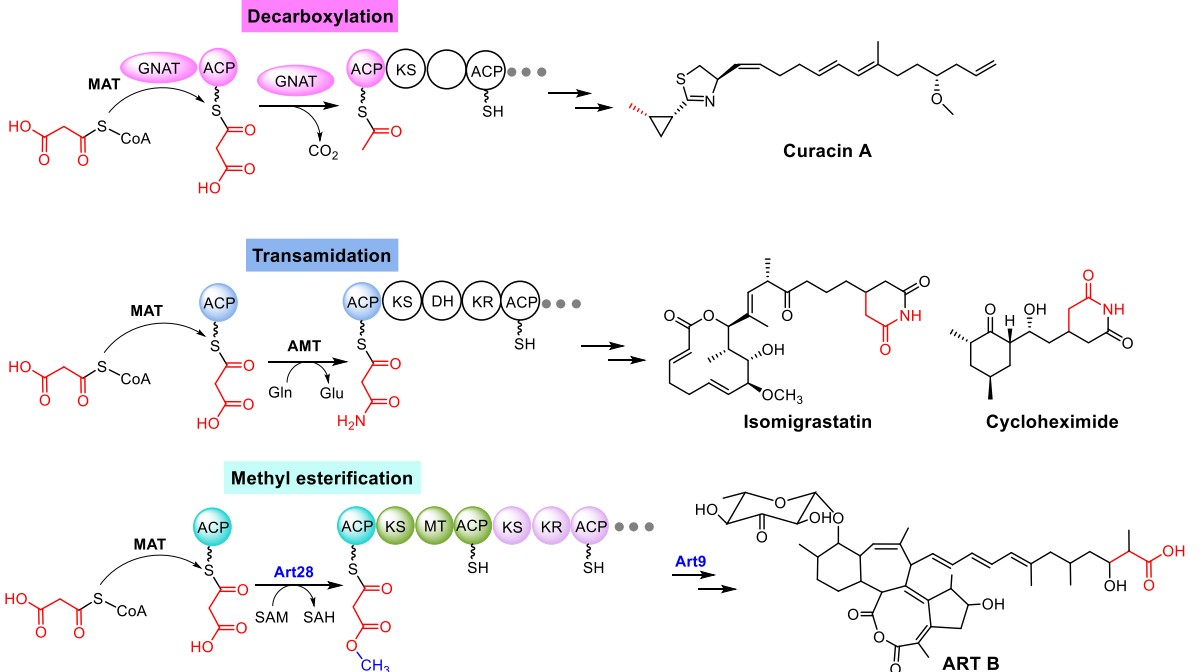

**Fig. 5** Selected initiation strategies involving on-line modification of the starter unit for modular PKSs[34–41]. **a** On-line modification initiation strategy for *cis*-AT PKSs. **b** On-line modification initiation strategies for *trans*-AT PKSs. ACP acyl carrier protein, AT acyltransferase, KS ketosynthase, KS$^Q$ decarboxylating ketosynthase, KR ketoreductase, DH dehydratase, MT methyltransferase, GNAT GCN5-related *N*-acetyltransferase, MAT malonyl-CoA: ACP acyltransferase, AMT amidotransferase.

peptidase XcmG upon secretion[43]. In addition, zeamines[44], colibactin[45], and zwittermycin[46] are also activated by post-assembly proteolytic processing; while calyculin[47], oleandomycin[48], and naphthyridinomycin[49] are activated by dephosphorylation, deglycosylation, and oxidative processes, respectively.

Methyl esterification of malonyl-ACP has also been observed in the biosynthesis of biotin. BioCs are isoenzymes of Art28, which hijack the fatty acid synthetic pathway by catalyzing methyl esterification of the malonyl group tethered to fatty acid ACPs[50]. As depicted in Supplementary Fig. 20, the biosynthesis of biotin also involves a methyl ester hydrolysis step catalyzed by BioH. Although both biotin and ART biosyntheses are initiated by methyl esterification of malonyl-ACP, they differ in two key respects: (i) the malonyl-ACP methyl ester is processed by non-iterative type I PKSs in ART biosynthesis but by iterative fatty acid synthases in biotin biosynthesis; (ii) Art9 catalyzes an off-line hydrolysis to activate ARTs, while BioH catalyzes an on-line reaction to generate an intermediate pimeloyl-ACP. The similarities between fatty acid synthesis and type II PKS systems suggest the possibility of methyl ester initiation logic in the latter, but we have not found Art28-Art9 homologs in type II PKS gene clusters. However, such systems may yet be discovered by genome

mining, while in the meantime, the results reported here should inspire synthetic biology efforts to generate novel polyketides incorporating terminal carboxylate groups[51,52].

## Methods

**Bacterial strains and plasmids.** All bacteria strains and plasmids used in this study are listed in Supplementary Table 9.

**DNA manipulation and sequence analysis.** General DNA manipulations were performed as described[53]. The primers used in this study (Supplementary Table 10) were synthesized by Generay (Shanghai, China). The PCR enzymes PrimeSTAR DNA polymerase (Takara, Japan) or Taq DNA polymerase (TransGene, Beijing, China) were used according to the manufacturers' instructions. Common DNA sequencing was performed by TianYi HuiYuan Co. (Beijing, China). Electro-poration of *B. subtilis* fmb60 was performed with an established protocol except the electric pulse condition was set as 15 kV/cm, 1 kΩ, and 25 μF[54]. Gene function annotations were performed with BLAST (http://www.ncbi.nlm.nih.gov/blast). Multiple alignments were carried out using CLUSTALW (https://www.ebi.ac.uk/Tools/msa/clustalw2/). Searching of the biosynthetic gene clusters containing *art28* homologs were performed with ClusterBlast of antiSMASH (https://antismash.secondarymetabolites.org/). The Art PKS domain organization and polyketide core structure were predicted by TransATor (https://transator.ethz.ch). Analysis of protein transmembrane region was performed with TMpred (https://embnet.vital-it.ch/software/TMPRED_form.html). Phylogenetic analysis was performed by MEGA 6.0.

**Construction of B. subtilis Δart11**. The *B. subtilis* Δ*art11* mutant was constructed by replacement of the *art11* gene with a tetracycline resistance cassette (*tet^R*) by homologous recombination. Briefly, the 1.1-kb upstream and 1.8-kb downstream fragments flanking *art11* gene were amplified by PCR with primer pairs 11-L-F/11-L-R and 11-R-F/11-R-R, respectively. The 1.8-kb *tet^R* cassette was amplified by PCR with primer pair tet^R-F/tet^R-R. The three fragments and the *Hind*III digested plasmid pRN5101[55] were stitched together using one-step cloning strategy to generate pRN5101::*art11*-UTD. Plasmid pRN5101::*art11*-UTD was then introduced into *B. subtilis* fmb60 via electroporation, and the tetracycline-resistant colonies that could grow up at 37 °C were selected as the *B. subtilis* Δ*art11* mutants, which were verified by PCR with primer pairs 11ver-F/Tetver-R and Tetver-F/11ver-R (Supplementary Fig. 21a). All above primers are listed in Supplementary Table 10.

**Construction of B. subtilis Δart(−1) and Δart(+1)**. *B. subtilis* Δ*art(−1)* and Δ*art(+1)* were constructed by single-crossover gene disruption. For the construction of *B. subtilis* Δ*art(−1)*, a 0.2-kb fragment from *art(−1)* was amplified by PCR with primer pair (−1)-F/(−1)-R using the genomic DNA of *B. subtilis* fmb60 as a template and cloned into the *Hind*III site of pRN5101 to generate pRN5101::*art(−1)*-S. Plasmid pRN5101::*art(−1)*-S was introduced into *B. subtilis* fmb60 via electroporation, and the erythromycin-resistant colonies that could grow up at 37 °C were selected as *B. subtilis* Δ*art(−1)* mutants, which were verified by PCR with primer pair −1ver-F/−1ver-R (Supplementary Fig. 21b).

*B. subtilis* Δ*art(+1)* was constructed in a similar manner. A 0.4-kb fragment from *art(+1)* was amplified by PCR with primer pair (+1)-F/(+1)-R and cloned into the *Hind*III site of pRN5101 to generate pRN5101::*art(+1)*-S. Successful construction of *B. subtilis* Δ*art(+1)* was verified by PCR with primer pair +1ver-F/+1ver-R (Supplementary Fig. 21c). All above primers are listed in Supplementary Table 10.

**Construction of B. subtilis Δart1, Δart28, Δart4, and Δart9**. The four mutants were constructed by in-frame deletion of each targeted gene. For the construction of *B. subtilis* Δ*art1*, two 1.0-kb DNA fragments flanking the *art1* gene were amplified with primer pairs 1-L-F/1-L-R and 1-R-F/1-R-R, respectively. The two fragments were inserted into the *Hind*III site of pRN5101 using one-step cloning strategy to generate plasmid pRN5101::*art1*-UD. After introduction of pRN5101::*art1*-UD into *B. subtilis* fmb60, the erythromycin-resistant transformants were selected as the single-crossover mutants, and then cultured in LB without antibiotics at 37 °C overnight to induce the second crossover. After that, the erythromycin sensitive colonies were screened out by replicating colonies on plates with or without erythromycin. The selected colonies were verified by PCR with primer pair 1-L-F/1-R-R as the desired *B. subtilis* Δ*art1* mutants (Supplementary Fig. 22a) and by sequencing the size-correct amplicons.

The other three *B. subtilis* fmb60 mutants were constructed in a similar way as that of *B. subtilis* Δ*art1*, except the primers used for cloning the two flanking DNA fragments of each gene and for verification of the mutant constructions were different. The 0.8-kb upstream and 0.8-kb downstream fragments flanking gene *art28* were amplified using primer pairs 28-L-F/28-L-R and 28-R-F/28-R-R, respectively. The 1.0-kb upstream and 1.1-kb downstream fragments flanking gene *art4* were amplified using primer pairs 4-L-F/4-L-R and 4-R-F/4-R-R, respectively. The 0.9-kb upstream and 0.8-kb downstream fragments flanking gene *art9* were amplified using primer pairs 9-L-F/9-L-R and 9-R-F/9-R-R, respectively. Successful constructions of *B. subtilis* Δ*art28*, Δ*art4*, and Δ*art9* were verified by PCR with primer pairs 28-L-F/28-R-R, 4-L-F/4-R-R, and 9-L-F/9-R-R, respectively (Supplementary Fig. 22b, c, d). Correct in-frame deletion of the targeted genes was validated by DNA sequencing. All above primers are listed in Supplementary Table 10.

**Site-directed mutagenesis of the MT domains of Art PKSs**. The conserved catalytic His residue of each MT domain of the Art PKSs (Art11, Art13, Art14, and Art17) was mutated to Ala by site-directed mutagenesis. For the MT domain of Art11, the 0.9-kb fragment upstream and 1.0-kb fragment downstream from the His$_{735}$ codon of *art11* (with 45 bp overlapping at the His$_{735}$ codon) were amplified with primer pairs 11MT-L-F/11MT-L-R and 11MT-R-F/11MT-R-R, respectively. In primers 11MT-L-R and 11MT-R-F, the His$_{735}$ codon (CAT) was replaced by an Ala codon (GCC). The two fragments were inserted into the *Hind*III site of pRN5101 using one-step cloning strategy to generate plasmid pRN5101::*art11MT*-PM, which was introduced into *B. subtilis* fmb60 via electroporation. The erythromycin-resistant transformants were selected as the single-crossover mutants, and cultured in LB without antibiotics at 37 °C overnight to induce the second crossover. The erythromycin sensitive colonies were screened out by replicating colonies on plates with or without erythromycin and were verified as the desired *B. subtilis* *art11MT** mutants by sequencing the fragments amplified by PCR with primer pair 11MT-L-F/11MT-R-R.

The other *B. subtilis* MT domain mutants were constructed in a similar manner, except the primers used for cloning the two fragments upstream and downstream of the His codons were different. The 0.9-kb upstream and 1.1-kb downstream fragments of His$_{3320}$ codon of *art13* (with 45 bp overlapping at the His$_{3320}$ codon) were amplified with primer pairs 13MT-L-F/13MT-L-R and 13MT-R-F/13MT-R-

R, respectively. The 0.9-kb upstream and 1.0-kb downstream fragments of His$_{1602}$ codon of *art14* (with 45 bp overlapping at the His$_{1602}$ codon) were amplified with primer pairs 14MT-L-F/14MT-L-R and 14MT-R-F/14MT-R-R, respectively. The 1.0-kb upstream and 1.1-kb downstream fragments of His$_{255}$ codon in *art17* (with 41 bp overlapping at the His$_{255}$ codon) were amplified using primer pairs 17MT-L-F/17MT-L-R and 17MT-R-F/17MT-R-R, respectively. The desired *B. subtilis* *art13MT**, *art14MT**, and *art17MT** mutants were verified by sequencing the fragments amplified by PCR with primer pairs 13MT-L-F/13MT-R-R, 14MT-L-F/14MT-R-R, and 17MT-L-F/17MT-R-R, respectively. All above primers are listed in Supplementary Table 10.

**Construction of the B. subtilis Δart28 complementation strains**. The 0.8-kb *art28* gene was PCR cloned from the genomic DNA of *B. subtilis* fmb60 with primer pair 28-F1/28-R1 (Supplementary Table 10) and inserted into the *Bam*HI and *Hind*III sites of plasmid pHY300$_{aprN}$[55] to generate pHY300$_{aprN}$::*art28*. The 0.8-kb codon-optimized *Bc-bioC* was synthesized by Generay (Shanghai, China) and inserted into the *Bam*HI and *Hind*III sites of plasmid pHY300$_{aprN}$ to afford pHY300$_{aprN}$::*Bc-bioC*. The complementation strains *B. subtilis*Δ*art28*/*art28* and Δ*art28*/*Bc-bioC* were obtained by introduction of pHY300$_{aprN}$::*art28* and pHY300$_{aprN}$::*Bc-bioC* into the Δ*art28* mutant via electroporation, respectively.

**Production of ARTs**. For the production of ART A-D and their congeners, *B. subtilis* fmb60, the *art* gene mutants, and the complementation strains were cultured in seed medium BPY (0.5% beef extract, 1.0% peptone, 0.5% yeast extract, 1% glucose, 0.5% NaCl, pH 7.0) as described[18]. The seed cultures were then inoculated (3% v/v) into Landy medium (4.2% glucose, 1.4% L-sodium glutamate, 0.05% MgSO$_4$, 0.05% KCl, 0.1% KH$_2$PO$_4$, 0.0015% FeSO$_4$, 0.005% MnSO$_4$, 0.0016% CuSO$_4$, pH 7.0) and cultured at 33 °C, 200 rpm for 36 h. After centrifugation, the supernatants were collected for isolation of ART A-D and their congeners.

**Isolation of compounds**. All ART compounds were isolated and prepared in a similar manner. Using the isolation of compound ART 9B as an example, the cell pellet of 60 L *B. subtilis* Δ*art9* culture broth was discarded after centrifugation and the supernatant was subjected to Diaion HP20 macroporous resin. After being eluted with methanol and concentrated at 25 °C in vacuo, the extract was subjected to a silica gel column (600 g, 200–300 mesh, 6.5 cm × 110 cm) and eluted sequentially with 4 L each of petroleum ether, petroleum-ethyl acetate (50/50, v/v), ethyl acetate, ethyl acetate-methanol (50/50, v/v), and methanol. The ethyl acetate fractions containing ART 9B were concentrated in vacuo and then fractionated on a reversed-phased C18 column (3.0 cm × 150 cm) eluted with acetonitrile/water gradient (40/60, 48/52, 75/25, and 90/10, v/v) sequentially. The fractions containing ART 9B were collected, concentrated in vacuo, and further fractionated by semi-preparative HPLC (Zorbax SB-C18, 5 µm, 9.4 mm × 250 mm, Agilent, CA, USA) eluted with acetonitrile/water (46/54, v/v) containing 0.1% formic acid at a flow rate of 3 mL/min to obtain 4.0 mg ART 9B.

ART 13B (2.5 mg) was obtained from 80 L *B. subtilis* Δ*art13MT** culture using the same procedure as that of ART 9B. The semi-preparative HPLC was performed with acetonitrile/water (47/53) containing 0.1% formic acid at a flow rate of 3.5 mL/min for the isolation of ART D and ART 11B; and 3.0 mg of each compound was obtained from 80 L *B. subtilis* Δ*art4* culture and 80 L *B. subtilis* Δ*art11MT** culture, respectively. ART 14A (1.0 mg) was isolated from 100 L *B. subtilis* Δ*art14MT** culture after refined by semi-preparative HPLC with acetonitrile/water (42/58) containing 0.1% formic acid at a flow rate of 3.0 mL/min.

**Isotope label feeding of B. subtilis fmb60 and the Δart9 mutant**. The seed cultures of *B. subtilis* strains were prepared as usual. Forty milligrams of [1-$^{13}$C] sodium acetate, [2-$^{13}$C] sodium acetate, or [$^{13}$CH$_3$]-methionine (filter-sterilized) was added to 1 L culture after culturing for 12 h, and again after 20 h in the fermentation medium. The fermentations were stopped after 36 h, and the supernatants were collected for compound isolation. The percentage of [$^{13}$C]-enrichment for each isotope labeling experiment was estimated by comparison of the $^{13}$C NMR signals with those of ART B and ART 9B at natural abundance.

**Construction of protein expression plasmids**. The 804-bp *art9* gene was cloned by PCR using the genomic DNA of *B. subtilis* fmb60 as a template with primer pair 28-9-F/28-9-R and inserted into the *Nde*I and *Bam*HI sites of pET28a to afford pET28a::*art9*. The 822-bp *art28* gene was cloned by PCR using primer pair 28-28-F/28-28-R and inserted into the *Nde*I and *Bam*HI sites of pET28a to afford pET28a::*art28*. The other five Art28 homologs (Clp905, Pas675, Thd235, Brf335, and Bac390) were synthesized and cloned into pET28a by Generay (Shanghai, China). The 237-bp fragment encoding the ACP domain of Art10 was cloned by PCR with primer pair 28-10-F/28-10-R and inserted into the *Nde*I and *Bam*HI sites of pET28a to afford pET28a::*art10acp*. All primers used to construct the expression plasmids are listed in Supplementary Table 10.

**Protein expression and purification**. *E. coli* BL21 (DE3) transformants harboring different protein expression plasmids were inoculated into LB with 50 µg/mL kanamycin and cultured at 37 °C, 220 rpm. The overnight culture was used to

inoculate LB with 50 μg/mL kanamycin at 1:100 dilution and incubated at 37 °C, 220 rpm until $OD_{600}$ reaching 0.6. Expression of N-His$_6$-tagged proteins were induced by the addition of isopropyl-$\beta$-thiogalactoside (IPTG) at a final concentration of 0.1 mM and cultured at 16 °C, 220 rpm for 18 h. To purify proteins, the cells were harvested by centrifugation and resuspended in lysis buffer (20 mM Tris-HCl, 500 mM NaCl, 5 mM imidazole, pH 7.9). After sonication, the cell debris was removed by centrifugation. The supernatant was loaded onto a Ni-NTA affinity column that had been equilibrated with lysis buffer, and then washed with washing buffer (20 mM Tris-HCl, 500 mM NaCl, 60 mM imidazole, pH 7.9) followed by elution buffer (20 mM Tris-HCl, 500 mM NaCl, 500 mM imidazole, pH 7.9). The desired fractions were combined, desalted using PD10 column, and concentrated by ultracentrifugation with an Amicon Ultra centrifugal filter (Millipore, MA, USA; molecular mass cutoff of 3 kDa for ACP$_{Art10}$ and 10 kDa for the other proteins). The proteins were stored at −80 °C in 25 mM HEPES buffer (pH 7.5) with 20% glycerol. Protein concentrations were measured by the Bradford assay[56].

**Art9 esterase assay.** The 50 μL reaction mixture consisted of 25 mM HEPES (pH 7.5), 1 mM ART 9B, and 50 μM Art9. The reaction was performed at 30 °C for 2 h before it was quenched by adding an equal volume of acetonitrile. After centrifugation, the supernatant was subjected to HPLC and MS analyses.

**Enzymatic assays of the malonyl-ACP O-MTs.** To prepare malonyl-ACP$_{Art10}$ as a substrate of Art28 and its homologs, purified apo-ACP$_{Art10}$ was treated with Sfp, a phosphopantetheinyl transferase from B. subtilis, as described[30]. The 200 μL reaction mixture consisting of 25 mM HEPES (pH 7.5), 10 mM MgCl$_2$, 5 μM Sfp, 0.1 mM ACP, and 0.5 mM malonyl-CoA was incubated at 30 °C for 1 h. The reaction mixture was then spun using a 0.5 mL Amicon Ultra centrifugal filter (molecular mass cutoff of 3 kDa) to collect ACP$_{Art10}$. After that, the collected ACP$_{Art10}$ was washed twice by 0.5 mL HEPES buffer (25 mM, pH 7.5) to remove malonyl-CoA. HPLC and LC-HRMS analyses showed that almost all apo-ACP$_{Art10}$ was converted to malonyl-ACP$_{Art10}$ by Sfp. The Art28 assay was set in 200 μL reaction mixture consisting of 25 mM HEPES (pH 7.5), 10 mM MgCl$_2$, 1 mM SAM, 10 mM NaCl, the collected malonyl-ACP$_{Art10}$ and 5 μM Art28. After incubation at 30 °C for 2 h, the reaction mixture was subjected directly to HPLC and LC-HRMS analyses. For detection of SAH production, the reaction was quenched by adding an equal volume of acetonitrile. After centrifugation, the supernatant was subjected to LC-MS analysis. The five Art28 homolog proteins were assayed with the same procedures.

To test whether Art28 can take malonyl-CoA as a substrate, the reaction was set in 200 μL HEPES buffer (25 mM, pH 7.5) consisting of 10 mM MgCl$_2$, 10 mM NaCl, 1 mM SAM, 0.5 mM malonyl-CoA, and 5 μM Art28. After incubation at 30 °C for 2 h, the reaction was quenched by adding an equal volume of acetonitrile and centrifuged. The supernatant was subjected to LC-MS analysis.

**Antibacterial assays.** The antibacterial assays of ARTs were performed with a micro-broth dilution method in the 96-well culture plate according to the Standard of National Committee for Clinical Laboratory[57]. Erythromycin was used as a positive control. The tested bacteria in this assay were incubated in Mueller-Hinton (MH) broth and normalized to an optical density of $10^6$ colony forming units (CFU)/mL. The detected substances were diluted using the double-dilution method in 96-well plates of MH broth. After incubation at 37 °C for 20 h, the minimal inhibitory concentration (MIC) was defined as the minimum concentration of the antibacterial substance at which no growth of the tested organisms could be detected.

**Cell membrane permeability assay.** The cell membrane permeability was tested with propidium iodide (J&K Scientific Ltd.) dye as described[58]. In brief, the overnight culture of Staphylococcus aureus ATCC 6538 was harvested and washed twice using buffer A (5 mM HEPES buffer, 5 mM glucose, pH 7.0). The cells were resuspended in the same buffer and adjusted to an optical density of 0.5. Subsequently, the aliquots of cell suspension (1 mL) were treated with ART B or ART 9B at 37 °C for 1 h. After propidium iodide (20 μM) was added, the cells were incubated for 20 min at 37 °C in dark, harvested by centrifugation, washed once, and resuspended in 1 mL buffer A for measuring fluorescence intensity spectrophotometrically in 96-well plate (at an excitation wavelength of 535 nm and an emission wavelength of 615 nm). Solvent DMSO was employed as a negative control; daptomycin (6 μM) + CaCl$_2$ (5 mM) was employed as a positive control.

**Spectroscopic analysis.** HPLC analyses of ARTs were carried out with an Apollo C18 column (5 μm, 4.6 mm × 250 mm, Alltech, IL, USA) on a Shimadzu HPLC system (Shimadzu, Kyoto, Japan). The column was developed with acetonitrile and water containing 0.1% formic acid at a flow rate of 0.8 mL/min. Percentage of acetonitrile was changed using the following gradient: 0–32 min, 30–65%; 32–33 min, 65–100%; 33–38 min, 100%; 38–39 min, 100–30%; 39–45 min, 30%. The detection wavelength was 280 nm.

HPLC analyses of the methylation of malonyl-ACP to malonyl-ACP methyl ester was performed using an Epic C18 column (5 μm, 4.6 mm × 250 mm, Alltech, IL, USA) on a Shimadzu HPLC system (Shimadzu, Kyoto, Japan). The column was developed with acetonitrile containing 0.1% trifluoroacetic acid and water containing 0.1% trifluoroacetic acid at a flow rate of 0.8 mL/min. Percentage of acetonitrile containing 0.1% trifluoroacetic acid was changed using the following gradient: 0–10 min, 36%; 10–45 min, 36–45%; 45–56 min, 45–90%; 57–67 min, 100%; 68–76 min, 36%. The detection wavelength was 210 nm.

LC-HRMS analyses and MS/MS analysis were performed on an Agilent 1260/6460 Triple-Quadrupole LC/MS system (Santa Clara, CA, USA) with the electrospray ionization source. LC-MS analysis of SAH was performed on an Agilent 1260 HPLC/6520 QTOF-MS system using a ZORBAX SB-AQ column (5 μm, 4.6 × 250 mm, Agilent Technologies, CA, USA). The column was developed with acetonitrile and water containing 0.1% formic acid at a flow rate of 0.8 mL/min. Percentage of acetonitrile was changed using the following gradient: 0–15 min, 0%; 15–30 min, 0–100%; 30–40 min, 100%; 40–55 min, 0%. Deconvoluted mass spectra of different ACP species were performed using an Orbitrap Fusion Tribrid mass spectrometer (Thermo Fisher Scientific, MA, USA). NMR spectra were recorded at room temperature on a Bruker Advance 500M NMR spectrometer (Billerica, MA, USA). MS and NMR data were analyzed using Qualitative Analysis B.07.00 and MestReNova 9.0.1, respectively.

Compound ART 11B: faint yellow amorphous powder; HRMS (ESI-TOF) m/z: $[M−H]^-$ Calcd for $C_{43}H_{58}O_{12}$, 765.3856; Found 765.3844, see Supplementary Fig. 1b; $^1H$ and $^{13}C$ NMR data, see Supplementary Table 2; $^1H$ and $^{13}C$ NMR, COSY, HSQC, and HMBC spectra, see Supplementary Fig. 1c–g.

Compound ART 13B: faint yellow amorphous powder; HRMS (ESI-TOF) m/z: $[M−H]^-$ Calcd for $C_{43}H_{58}O_{12}$, 765.3856; Found 765.3833, see Supplementary Fig. 2b; $^1H$ and $^{13}C$ NMR data, see Supplementary Table 3; $^1H$ and $^{13}C$ NMR, COSY, and HSQC spectra, see Supplementary Fig. 2c–f.

Compound ART 14A: faint yellow amorphous powder; HRMS (ESI-TOF) m/z: $[M+H]^+$ Calcd for $C_{37}H_{50}O_8$, 623.3578; Found 623.3586, see Supplementary Fig. 3b; $^1H$ NMR data, see Supplementary Table 4; $^1H$ NMR and COSY spectra, see Supplementary Fig. 3c and d; MS/MS analysis of ART 14A, see Supplementary Fig. 4.

Compound ART D: faint yellow amorphous powder; HRMS (ESI-TOF) m/z: $[M+H]^+$ Calcd for $C_{43}H_{58}O_{12}$, 767.4001; Found 767.4011, see Supplementary Fig. 7b; $^1H$ and $^{13}C$ NMR data, see Supplementary Table 5; $^1H$ and $^{13}C$ NMR, COSY, and HMBC spectra, see Supplementary Fig. 7c–f.

Compound ART 9B: faint yellow amorphous powder; HRMS (ESI-TOF) m/z: $[M+H]^+$ Calcd for $C_{45}H_{62}O_{12}$, 795.4314; Found 795.4315, see Supplementary Fig. 8b; $^1H$ and $^{13}C$ NMR data, see Supplementary Table 6; $^1H$ and $^{13}C$ NMR and HMBC spectra, see Supplementary Fig. 8c–e.

**Reporting summary.** Further information on research design is available in the Nature Research Reporting Summary linked to this article.

## Data availability

The genome information of B. subtilis fmb60 is publicly available through NCBI under accession number NZ_LYMC01000002.1. The antiSMASH database is publicly available [https://antismash-db.secondarymetabolites.org/]. The authors declare that all data supporting the findings of this study are available within the article and its Supplementary Information file. Source data are provided with this paper.

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

## Acknowledgements

We thank Dr. Jinwei Ren, Wenzhao Wang, and Jingfang Liu, Institute of Microbiology, CAS, for technical supports. This work was supported in part by grants from MOST of China (2018YFA0901900), China NSFC (32000042 and 32025002), and Joint project of CAS-Inner Mongolia (KEXUE2019GZ05).

## Author contributions

Y.C. and Z.L. conceived the project. Y.C., P.L. and G.P.H. wrote the paper. P.L. and Y.Z. performed the sequence analyses. P.L., W.T. and J.Z. constructed and analyzed the mutant strains. P.L., Z.G. and M.W. identified the ART congeners. P.L. and M.C. conducted the in vitro enzymatic studies.

## Competing interests

The authors declare no competing interests.
