## [Peer Review File · Nature Communications]

Reviewers' Comments:

Reviewer #1:

Remarks to the Author:

The building blocks employed to initiate polyketide biosynthesis represent an important source of structural diversity in these natural products. In addition, as shown here for the first time by analysis of aurantinin biosynthesis, they can be leveraged to introduce protecting group chemistry into the pathway, which is followed by late-stage unmasking to yield the antibiotic product. This elegant conclusion is supported by generation and analysis of a battery of genetic mutants, as well as reconstruction *in vitro* of the O-methylation chemistry using purified enzymes, with both types of approach supported by analytical chemistry (HPLC-MS and NMR). While the results described here are novel and will be of substantial interest to the biosynthetic community, there are a number of points that need to be addressed before the ms becomes suitable for publication in *Nat. Commun.* These are detailed below.

Major comments:

1. The biggest concern are the results of the inactivation mutants (as shown for example in Fig. 1, c and d). The wild type chromatogram exhibits multiple peaks, only some of which can be attributed to mature ART metabolites (A, B and D), while other peaks are also present. However, in the *art11*, *art1*, *art28* and *art17MT** mutants, essentially ALL peaks have disappeared. Do the authors have an explanation for this finding? Also, in the wild type chromatogram, where is ART C?
2. Page 5, line 82. It is rather unreasonable to state that analysis of the trans-AT PKS provided 'few clues' as to the origin of the two polyketide chains, as there is publicly-available software (TransATor, doi: 10.1038/s41589-019-0313-7; <https://github.com/pcm32/transator-container>) which is capable not only of analyzing the composition in domains of trans-AT PKS subunits, but of predicting the structure of the resulting intermediates with good confidence. What is the result of analysis using TransATor, as this could provide important validation for the tool, or rather highlight certain deficiencies of interest to the community?
3. Page 8, line 156. Did the authors consider that the 'unknown' ca. 100 residue domain C-terminal to the ACP in *Art10* might be a docking domain? Such elements have been identified in trans-AT PKS systems (doi: 10.1021/jacs.5b13372; doi: 10.1021/acschembio.6b00345; doi: 10.1016/j.jsb.2020.107581), and could mediate the necessary specific communication between *Art10* and *Art11*. This region should be reanalyzed in this context, as well as that of the N-terminus of *Art11* for a partner docking domain.
4. Page 12. As this is not the first time that a 'protecting group strategy' has been described for modular natural product biosynthesis, the authors should reference prior art in this part of the discussion (e.g. doi: 10.1038/nchembio.688 and subsequent articles which cite it (doi: 10.1039/c4sc01927j , etc.))
5. Supplementary Fig. 2. It appears that the two large branches of this tree have been artificially superimposed on each other (a vertical line from the green branch overlaps with a horizontal line from the blue branch). Is there a benign explanation for this observation?
6. Supplementary Fig. 8. It is remarkable that all of these reactions went to 100%, despite the non-native nature of the ACP domain to which the substrate was tethered. The authors might like to comment on this result, and the fact that it implies that protein-protein interactions contributing to specificity are limited in this case.

More minor comments:

1. Page 3, Introduction. The authors should better describe in their introduction the basic function of polyketide synthases (including the roles of the key domains, defining what 'modules' are, etc.), as well as introducing the distinction between cis-AT and trans-AT PKSs. This is important information for understanding the remainder of the paper (including several figures), in which a basic knowledge of PKS biochemistry is assumed.
2. Page 3, line 32: suggest, 'analogous to fatty acids'
3. Page 3, line 33. It is too simplistic to state that extender units are 'malonyl-CoA' as a variety of monomers are employed (doi: 10.1039/c5np00112a).
4. Page 3, lines 33–35. Suggest that the authors find an alternative to the words 'diversity' and 'diverse' which are repeated 3 times.
5. Page 3, line 47. The authors describe the apparent origin of the short polyketide chain as 'extraordinary' because if it were generated by classical PKS chemistry, it would require a tail-to-

tail condensation. However, as alternative origins are plausible (and indeed they show that it arises from intact incorporation of succinate), the 'extraordinary' nature of this chain is rather exaggerated.

6. Page 4, line 70. Citations should be inserted here referencing the original description of trans-AT PKS systems by the groups of Piel (doi: 10.1073/pnas.222481399) and Shen (doi: 10.1073/pnas.0537286100).

7. Fig. 1. For clarity, the set of chromatograms on the right should be labeled as panel d.

8. Page 5, line 88. The β -methylation cassette typically includes a discrete ACP domain, and indeed, the authors identified Art7 as a stand-alone ACP. It should thus be mentioned here. (Similarly, page 6, line 99, should be 'five genes')

9. Page 5, line 94. Reference should be made at this point to the appropriate Supplementary figures/tables, which should be reordered to be sequential. Also, the authors should comment on the yields, as PKS mutagenesis is typically associated with substantial reductions in titers.

10. Page 6, line 98. Non-expert readers will not understand what the 'co-linearity rule' refers to. An explanation should therefore be provided (and could be included in the revised introduction).

11. Page 6, line 103. A more accurate description is that the β -branching enzymes together incorporate the methyl groups into the intermediates generated on Art11.

12. Page 6, lines 105 and 109. It would be better to say that they 'hypothesized' that Art17 assembled the short polyketide chain (but again, this would have been pre-validated by the use of TransATor), and Art2 initiates the biosynthesis

13. Page 6, line 110. The authors could already point out here that use of succinate as a starter unit would explain the apparently 'extraordinary' tail-to-tail condensation of two acetate units referred to earlier in the text.

14. Fig. 2. The authors should explain the two colors of ACP in the β -methylation reaction inset (i.e. that the reaction sequence occurs in two different modules of Art11). In addition, for the non-expert reader, the legend should include definitions of all domain name abbreviations (ACP, KS, KSo, etc.). The authors should also explicitly state what the colored carbons refer to (i.e. points at which the methyl groups have been removed by inactivation of the corresponding cMT domains)

15. Page 7, line 122. It would be better to write that the feeding experiments 'suggested a possible origin for the free terminal carboxylic acid group' as being via oxidative cleavage, as the authors go on to show that it is generated by an alternative route (i.e. 'indicated' is misleading)

16. Page 7, line 125 (also Page 7, lines 135 and 139 and Page 8, line 145). The polyketide chain is not strictly-speaking 'split', but the BVMOs rather insert oxygen into the chain.

17. Page 7, line 130. Again, appropriate reference needs to be made to the Supplementary materials.

18. Page 7, lines 120–134. Suggest using a synonym to 'expose' to avoid repetition, for example, 'reveal' or 'liberate'

19. Page 8, line 142. But saying that this particular experiment was done 'carefully', this would imply that the other experiments to which they refer (and by other researchers) were not.

20. Page 8, line 145. Suggest 'established' rather than 'shaped'

21. Page 9, line 171. Replace 'resulted in' with 'yielded'

22. Page 9, line 175. By 'typical type I PKS gene cluster' do they mean 'cis-AT PKS'? Again, this illustrates why more care needs to be taken to introduce the two large classes of modular PKSs from the beginning of the ms. In terms of the hybrid PKS-NRPS systems, do these incorporate cis-AT or trans-AT PKS components?

23. Page 10, line 192. Suggest, 'The obtained data clearly show that ARTs only inhibit growth of Gram-positive...'

24. Figure 5. In this figure (and in the accompanying text, page 12), the authors have mixed chain initiation strategies for both cis-AT and trans-AT PKSs. In light of the fact that the two systems apparently have distinct evolutionary origins which likely contribute to the observed diversity, it would be appropriate to separate the strategies by PKS family. Again, definitions of domain/stand-alone enzyme acronyms should be provided. Also, 'loading starter unit onto an ACP'; 'by on-line modification of the starter unit'. Capitalize Transamidation, but not Type I. Also suggest as a title, 'Selected initiation strategies...', as this is not a complete list. Finally, citations to the relevant work demonstrating these strategies should be added.

25. Page 10, line 199. Suggest, '...build on this work to establish a common...'

26. Page 10, line 201. Having introduced the concept of trans-AT PKSs earlier, it is not necessary to use the terms here (and indeed, the addition of a second one ('AT-less') will be confusing at this stage)

27. Page 13, line 258. Suggest, 'However, such systems may yet be discovered by genome mining, while in the meantime, the results reported here should inspire synthetic biology efforts to generate novel polyketides incorporating terminal carboxylate groups.' (And here, a reference to a review on PKS genetic engineering would be appropriate)
28. Page 20, line 460. There is an odd mixture here of cloning methodology, and protein expression/purification. These should be separated, and reference made to the appropriate table in the Supplementary.
29. Supplementary Table 1. The listed domain composition of Art11 isn't consistent with Figure 2, as according to the figure, module 4 contains a tandem of ACP domains. Also, the authors could distinguish between the ECH1 and ECH2 homologs (Art20 and Art21).
30. Supplementary Fig. 6. Can the authors explain why the difference between the observed and calculated masses is larger in part a than in part b, for the same species?
31. Supplementary Fig. 7. Labeling should be added to this figure to identify to which of the multiple species potentially present in each assay the observed HPLC-MS peaks correspond. The legend to a is also incorrect, as in fact, malonyl-CoA methyl ester wasn't detected, but its expected mass is indicated. Same comment for part b of the legend.

Typos/grammatical errors

1. Page 4, line 61: 'are reminiscent'
2. Page 5, line 85 (and Page 8, line 161). S-adenosyl-methionine (adenosyl not capitalized)
3. Page 6, line 116. 'Overall, the incorporation of all seven...'
4. Page 7, line 124. 'monooxygenases' (plural)
5. Page 7, line 135. Oocydin not Occydin
6. Page 8, line 140. , not ;
7. Page 9, line 181. 'from the other MTs'
8. Page 10, line 185. 'All five proteins...'
9. Page 10, line 205. 'approximately' instead of 'about'
10. Page 10, line 205. 'and Art14KS4 contains an Asn in place of the first His...'
11. Page 11, line 207. 'have been observed'
12. Page 11, line 207. 'We therefore propose that...'
13. Page 11, line 215. Remove 'is' before 'originates'
14. Page 12, line 229. 'such as isomigrastatin and cycloheximide, although this mechanism requires further validation by in vitro studies.'
15. Page 12, line 232. 'and in the case of the ARTs,'
16. Page 12, line 237. 'Art28-Art9-type' (as the exploration is not limited to this specific pair)
17. Page 12, line 240. 'inhibitory activity'

(A few comments on the Materials and Methods, but the whole section needs careful proof-reading)

18. Page 15, line 350. 'are listed'
19. Page 15, line 256. Remove 'a' before 1.1 kb
20. Page 15, line 365. Appropriate reference should be made to a table in the Supplementary for the primer sequences (and the same comment for the subsequent sections).
21. Page 18, line 404. 'downstream from the His735'
22. Supplementary Fig. 1 legend, line 101. 'are highlighted'

Reviewer #2:

Remarks to the Author:

Li et al present their most recent findings on the biosynthesis of aurantinin and its congeners. The compounds and their biogenesis has been under investigation since the late 1970s and its biosynthesis has been (vaguely) elucidated before (citation 12, <https://doi.org/10.1021/acs.jafc.6b04455>) using genome analysis.

The current ms. tackles the assignment of the different (cryptic) ORF to specific functions in the biosynthesis and elucidate some interesting features. Apart from the possible (but not yet proven!) convergent biosynthetic layout (as also found in a number of other polyketides), the authors report one to the best of my knowledge new function: The loading unit malonic acid is selectively

methylated by an methyltransferase which is essential for the activity of the whole PKS assembly line. Its significance for the self-protection of *B. subtilis* fmb609 is discussed.

The authors study this methyltransferase in vivo and to some extent in vitro and identify some homologs in other PKS gene clusters.

Indeed, the biosynthetic mechanism is interesting and worth publication. However, for publication in a high-impact journal more definitive answers to some pressing questions would be necessary. The authors themselves suggest some further experiments that would bolster up this study in the discussion. Thus, this article largely rests on the on-line methyl esterification initiation strategy, which indeed is unprecedented but at the same time rather a curiosity than a mechanism of great relevance for natural product research.

The more general message of this ms. thus is that researchers should look at least twice at unusual ORF – in vivo and in vitro.

Overall, I recommend submission to a more specialized journal.

Reviewer #3:

Remarks to the Author:

The manuscript by Li et al. describes a partial dissection of the Aurantinins (ARTs), which are antibacterial natural products generated by a trans-AT modular polyketide synthase system. The authors conducted a series of gene disruption (and allied complementation) experiments to determine the boundaries of the art biosynthetic gene cluster (BGC) that is comprised of 28 genes across ~80 kb. They also performed precursor incorporation studies to identify the subunits and source of methyl groups within the pathway. The aurantinins are structurally unique metabolites that contain a 6/7/8/5 fused ring system. Two hypotheses currently drive the studies to test whether a single linear polyketide chain is generated that subsequently forms the unique tetracyclic ring system, or two linear chains converge to create the core molecule. Figure 1 provides the data relating to the outer boundaries of the BGC, and a series of comprehensive gene deletion and complementation experiments were conducted to support their characterization. A series of new molecules that represent intermediates in the pathway from gene deletion studies were isolated and characterized by MS and various rigorous NMR methods. The authors made the surprising finding that the starter unit of the pathway is in fact malonyl-CoA methyl ester, which they describe as a “protecting group” that renders the molecule inactive until that late stage of biosynthesis where the methyl group is removed by an esterase to unveil the carboxylate. It is this form of the molecule (ART B) that displays antibiotic activity against Gram positive bacteria. Biochemical studies were conducted on the recombinantly expressed methyltransferase (ART28) to demonstrate directly the formation of malonyl-CoA methyl ester. Direct studies were also performed on the corresponding esterase (ART9) to generate ART B (maximally active) from ART 9B (inactive).

There is significant merit in this study. The finding of a new type of starter unit that appears to “protect” the organism from its own antibiotic (self-resistance?) is important. Using in silico mining of natural product BGCs, the authors were able to identify numerous systems that appear to employ a similar biosynthetic strategy. This aspect of the work is perhaps the single most important contribution. Other aspects of the study (BGC sequencing, gene deletion, complementation) are relatively standard methods, but the authors were able to dissect some important aspects of the pathway.

Major Criticisms: On the other hand, there are numerous aspects of the work that fall short. First, the most significant question is arguably the mode of assembly of the unusual 6/7/8/5 tetracyclic ring system. Although the authors present this as a challenge, the work leaves this question untouched. Second, although the hypothesis is presented that the malonyl-CoA methyl ester starter provides some form of self-protection to the producing microbe, the authors fail to address how this “prodrug” strategy operates at the molecule level on the proposed cellular target (see reference 12), or whether other types of cell targets may be involved in the antibacterial activity. In this respect, the authors' scholarship on this topic is inadequate. What other biosynthetic systems employ transient chemical modification as a mode of self-protection? Indeed, there are numerous examples in the literature and the authors need to cover this topic in the discussion section to enhance scholarship and to place their work in a proper perspective.

Minor Criticisms: The field has settled on the term "trans-AT" PKS as opposed to "AT-less". On line 36-37 the authors make the unsupported statement that the most common PKS starter unit is malonyl-CoA. How do they know this?

Reviewer 1's remarks:

The building blocks employed to initiate polyketide biosynthesis represent an important source of structural diversity in these natural products. In addition, as shown here for the first time by analysis of aurantinin biosynthesis, they can be leveraged to introduce protecting group chemistry into the pathway, which is followed by late-stage unmasking to yield the antibiotic product. This elegant conclusion is supported by generation and analysis of a battery of genetic mutants, as well as reconstruction in vitro of the O-methylation chemistry using purified enzymes, with both types of approach supported by analytical chemistry (HPLC-MS and NMR). While the results described here are novel and will be of substantial interest to the biosynthetic community, there are a number of points that need to be addressed before the ms becomes suitable for publication in *Nat. Commun.* These are detailed below.

Major comments:

1. The biggest concern are the results of the inactivation mutants (as shown for example in Fig. 1, c and d). The wild type chromatogram exhibits multiple peaks, only some of which can be attributed to mature ART metabolites (A, B and D), while other peaks are also present. However, in the art11, art1, art28 and art17MT* mutants, essentially ALL peaks have disappeared. Do the authors have an explanation for this finding? Also, in the wild type chromatogram, where is ART C?

Response: *Thanks for the reviewer's careful inspections. The multiple peaks in the HPLC chromatogram of wild type strain are inferred to be aurantinin (ART) analogues based on their characteristic UV-vis spectrum of the unique triene moiety in ARTs and LC-MS data. During the isolation process, we noticed that some of the ART analogues are not stable, which may be one of the reasons that only ART A-D were identified till now.*

In the art gene mutant strains, if the inactivated gene is essential to the ART polyketide chain assembly, the production of all those ART analogues will be blocked and those multiple peaks will disappear. For the four mutant strains that the reviewer noticed, (i) genes art11 and art17 encode ART polyketide synthases that are essential for the assembly of the polyketide chain; (ii) the art28 gene was proposed to be involved in the polyketide chain initiation process by us in this manuscript; and (iii) the art1 gene encodes a conserved anti-terminator that is responsible for regulatory control the transcription of the art biosynthetic gene cluster. Once gene art1 was inactivated, the whole art gene cluster failed to work, leading to no ARTs production. Similar regulatory manners existed in several cases of secondary metabolites from Firmicutes (DOI: 10.1038/nmicrobiol.2017.3). In summary, all these four genes are either structure genes essential for the ART polyketide chain assembly or polyketide genes positively controlling the ART gene cluster. Therefore, all peaks related to ARTs have completely disappeared in these four mutants.

In addition, the peak for ART C was marked on HPLC in the revised manuscript based on high-resolution mass analysis and comparison with the reported data.

2. Page 5, line 82. It is rather unreasonable to state that analysis of the trans-AT PKS provided 'few clues' as to the origin of the two polyketide chains, as there is publicly-available software (TransATor, doi: 10.1038/s41589-019-0313-7; <https://github.com/pcm32/transator-container>) which is capable not only of analyzing the composition in domains of trans-AT PKS subunits, but of predicting the structure of the resulting intermediates with good confidence. What is the result of analysis using TransATor, as this could provide important validation for the tool, or rather highlight certain deficiencies of interest to the community?

Response: *As the reviewer suggested, we tested several different trans-AT PKS clusters on TransATor including the ART biosynthetic gene cluster. This publicly available analysis tool is user-friendly and highly reliable in prediction of biosynthetic domains of trans-AT PKS clusters. The predicted structures by TransATor are informative in all tested*

cases.

The TransATor results about the ART biosynthetic gene cluster were shown below. As we can see, the resulting polyketide intermediate is similar to the proposed core polyketide chain of ART, but provides limited information regarding the special starter unit, malonyl-ACP methyl ester, and the uncommon succinyl-CoA unit. Considering the spread of these two uncommon units, we believe the accuracy of *in silico* tools like TransATor could be further increased by incorporating the knowledge about malonyl-ACP methyl ester starter unit and succinyl unit described in this manuscript.

Proposed polyketide intermediate of ART in S-Fig. 16

Predicted polyketide intermediate of ART by TransATor

The bioinformatic analysis of the ART gene cluster by tools like transATor was added in the revised manuscript as: 'In silico analysis results of the art gene cluster using bioinformatic tools like TransATor¹⁷ were informative but provided few clues regarding the two-polyketide chain assembly model and the unusual starter units of the polyketide chains.' (page 5, line 88)

3. Page 8, line 156. Did the authors consider that the 'unknown' ca. 100 residue domain C-terminal to the ACP in Art10 might be a docking domain? Such elements have been identified in trans-AT PKS systems (doi: 10.1021/jacs.5b13372; doi: 10.1021/acscchembio.6b00345; doi: 10.1016/j.jsb.2020.107581), and could mediate the necessary specific communication between Art10 and Art11. This region should be reanalyzed in this context, as well as that of the N-terminus of Art11 for a partner docking domain.

Response: The C-terminal domain of Art10 and the N-terminal incomplete KS domain of Art11 were aligned with the docking domains from the mentioned references and others. However, neither the C-terminal domain of Art10 nor the N-terminal domain of Art11 contain a docking domain for protein-protein communication.

In addition, when we analyzed with TransATor, it was showed that the first KS domain is consisted of the C-terminal domain of Art10 and the N-terminal incomplete KS domain of Art11, indicating that Art10 and Art11 is one protein. To exclude a mistake introduced by sequencing, the DNA fragment spanning the interspace of genes art10 and art11 was PCR amplified and the resulting amplicon was sequenced again. The sequencing result showed that the original sequence is correct and the stop codon is definitely existed between these two genes. Therefore, we propose that this split KS domain is nonfunctional and could play a transferring role during ARTs biosynthesis.

4. Page 12. As this is not the first time that a 'protecting group strategy' has been described for modular natural product biosynthesis, the authors should reference prior art in this part of the discussion (e.g. doi: 10.1038/nchembio.688 and subsequent articles which cite it (doi: 10.1039/c4sc01927j , etc.))

Response: The content about protecting group strategy in natural product biosynthesis was added to the discussion section as suggested. The related papers were also referenced in the revised manuscript.

The added content in the revised manuscript is attached as below (page 13, line 257):

‘Similar protecting group strategy has been observed during the biosynthesis of a number of natural products and a versatile range of activation strategies have been recruited. For example, the ‘pro-drug’ of antibiotic xenocoumacin is activated via cleavage of an acylated D-asparagine by a membrane-anchored peptidase XcmG upon secretion⁴². In addition, zeamines⁴³, colibactin⁴⁴, and zwittermycin⁴⁵ are also activated by post-assembly proteolytic processing; while calyculin⁴⁶, oleandomycin⁴⁷, and naphthyridinomycin⁴⁸ are activated by dephosphorylation, deglycosylation, and oxidative processes, respectively.’

5. Supplementary Fig. 2. It appears that the two large branches of this tree have been artificially superimposed on each other (a vertical line from the green branch overlaps with a horizontal line from the blue branch). Is there a benign explanation for this observation?

Response: We greatly appreciated reviewer’s careful inspection on this figure! It was overlooked by us that the phylogenetic analysis figure generated from the website had something wrong due to the display problem. We are really sorry about that! To solve this problem, the raw data of the phylogenetic analysis was retrieved and viewed with another viewer software (TreeDyn). This new figure was used in the revised Supplementary Information as reordered Supplementary Fig. 6 (as shown below).

6. Supplementary Fig. 8. It is remarkable that all of these reactions went to 100%, despite the non-native nature of the ACP domain to which the substrate was tethered. The authors might like to comment on this result, and the fact that it implies that protein-protein interactions contributing to specificity are limited in this case.

Response: Actually, the initial reactions were not as efficient as those showed in Supplementary Fig. 14 (the original Supplementary Fig. 8), but rather with different efficiencies as shown below. After further optimization of the reaction conditions, including substrate purity, buffer pH adjustment, enzyme concentration, and reaction time, these reactions were ultimately optimized as shown in the manuscript.

Initial assay conditions: the 200 μ L reaction mixture consisting of 25 mM HEPES (pH 8.0), 10 mM $MgCl_2$, 1 mM SAM, 10 mM NaCl, the collected malonyl-ACP_{Art10} and 1 μ M Art28 was incubated at 30 °C for 1 hour.

On the other hand, we performed the homologous analysis of these different methyltransferase from various biosynthetic gene clusters, they all shared pretty high identities with Art28 (Clp905: 61.1%; Pas675:52.0%; Brf335:60.2%; Thd235:72.3%; Bac390: 61.6%), indicating that these methyltransferases are well conserved.

More minor comments:

1. Page 3, Introduction. The authors should better describe in their introduction the basic function of polyketide synthases (including the roles of the key domains, defining what ‘modules’ are, etc.), as well as introducing the distinction between cis-AT and trans-AT PKSs. This is important information for understanding the remainder of the paper (including several figures), in which a basic knowledge of PKS biochemistry is assumed.

Response: Thanks for the reviewer’s excellent suggestion! The more detailed narrative about functions of modular polyketide synthases was added in Introduction of the revised manuscript. Trans-AT PKS was also described in this part.

The first paragraph was changed as below (page 3, line 31):

‘Polyketides are a large family of structurally diverse natural products with varied biological and pharmacological activities including antibacterial, antitumor, and immunosuppressant^{1,2}. A great number of polyketides are constructed by giant modular polyketides synthases (PKSs) with multifunctional domains. After initiated by the loading module, the polyketide chain will be elongated iteratively by each extension module in a manner analogous to fatty acid biosynthesis. The acyltransferase (AT) domain loads a specific extender unit to acyl carrier protein (ACP) and the ketosynthase (KS) domain performs a two-carbon addition to ACP-tethered acyl-thioesters via head-to-tail Claisen condensation. The newly synthesized β -keto thioester intermediates could be modified by additional domains (e.g. ketoreductase (KR), dehydratase (DH), enoyl reductase (ER), and methyltransferase (MT)) in the extension module

before they are released by an offloading module which usually contains a thioesterase (TE) or a reductase (R) domain³⁻⁵. Specifically, a large family of modular PKSs is *trans*-AT (or AT-less) PKS, which does not have an AT domain in each extension module as *cis*-AT PKS but shares a standalone AT by multiple extension modules⁶⁻⁸. In principle, the polyketide core structures are consistent with the domain organizations of module PKSs, which was called the co-linearity rule⁹. The vast structural diversity of polyketides is introduced by different combinations of initiation, extension, and termination steps as well as versatile tailoring processes. For the initiation process, malonyl-coenzyme A (CoA) is the most frequently used starter unit, which is usually decarboxylated to an acetyl-thioester to begin the extension process. Consequently, most polyketides constructed in this way have a terminal methyl group unless it is modified by tailoring enzymes¹⁰⁻¹².

2. Page 3, line 32: suggest, ‘analogous to fatty acids’

Response: *This sentence was deleted.*

3. Page 3, line 33. It is too simplistic to state that extender units are ‘malonyl-CoA’ as a variety of monomers are employed (doi: 10.1039/c5np00112a).

Response: *Thanks for pointing out this improper statement! Since there are increasingly newly identified extender units besides malonyl-CoA, such as various CoA or ACP-linked extender units, we deleted ‘malonyl extender units’ in the revised manuscript.*

4. Page 3, lines 33–35. Suggest that the authors find an alternative to the words ‘diversity’ and ‘diverse’ which are repeated 3 times.

Response: *We deleted a ‘diverse’ in the revised manuscript.*

5. Page 3, line 47. The authors describe the apparent origin of the short polyketide chain as ‘extraordinary’ because if it were generated by classical PKS chemistry, it would require a tail-to-tail condensation. However, as alternative origins are plausible (and indeed they show that it arises from intact incorporation of succinate), the ‘extraordinary’ nature of this chain is rather exaggerated.

Response: *‘Extraordinary’ was replaced by ‘uncommon’ in the revised manuscript.*

6. Page 4, line 70. Citations should be inserted here referencing the original description of *trans*-AT PKS systems by the groups of Piel (doi: 10.1073/pnas.222481399) and Shen (doi: 10.1073/pnas.0537286100).

Response: *The references were cited as ref. 6 and 7.*

7. Fig. 1. For clarity, the set of chromatograms on the right should be labeled as panel d.

Response: *Figure 1 was revised as suggested and the main peaks of ART analogues on the chromatograms were labeled.*

8. Page 5, line 88. The β -methylation cassette typically includes a discrete ACP domain, and indeed, the authors identified Art7 as a stand-alone ACP. It should thus be mentioned here. (Similarly, page 6, line 99, should be ‘five genes’)

Response: *The art7 gene which encodes the stand-alone ACP domain was included to the β -branching genes. And, the*

'four genes' for β -branching was changed to 'five genes' in revised manuscript. This change was made in the following text as well.

9. Page 5, line 94. Reference should be made at this point to the appropriate Supplementary figures/tables, which should be reordered to be sequential. Also, the authors should comment on the yields, as PKS mutagenesis is typically associated with substantial reductions in titers.

Response: *The related Supplementary figures and tables were referred (page 6, line 101) and their numbers were reordered sequentially both in the revised manuscript and in Supplementary information.*

The comment 'Productions of all the demethylated ART congeners were reduced dramatically in those mutant strains with PKS mutagenesis' was added in the revised manuscript (page 6, line 104).

In addition, the yields of different demethylation congeners produced by MT mutants were also described in the part of Isolation of compounds in Materials and methods, respectively (page 18, line 374).

10. Page 6, line 98. Non-expert readers will not understand what the 'co-linearity rule' refers to. An explanation should therefore be provided (and could be included in the revised introduction).

Response: *The concept of co-linearity rule for PKS was provided in the first paragraph of the revised manuscript (page 3, line 42):*

'In principle, the polyketide core structures are consistent with the domain organizations of module PKSs, which was called the co-linearity rule⁹.'

11. Page 6, line 103. A more accurate description is that the β -branching enzymes together incorporate the methyl groups into the intermediates generated on Art11.

Response: *We rephrased the sentence as 'implying that the methyl groups at C-5 and C-7 are incorporated into the intermediates generated on Art11 by the β -branching enzymes together' (page 6, line 110).*

12. Page 6, lines 105 and 109. It would be better to say that they 'hypothesized' that Art17 assembled the short polyketide chain (but again, this would have been pre-validated by the use of TransATor), and Art2 initiates the biosynthesis.

Response: *We have predicted the structure using art gene cluster as the query by TransATor. The generated results suggested that Art17 could be responsible for the short polyketide chain assembly despite not inclusive of the special succinate unit. Consequently, we replaced the word 'proposed' with 'hypothesized' in the revised manuscript.*

13. Page 6, line 110. The authors could already point out here that use of succinate as a starter unit would explain the apparently 'extraordinary' tail-to-tail condensation of two acetate units referred to earlier in the text.

Response: *We rephrased the sentence as 'by loading an uncommon starter unit succinyl to the first ACP of Art17' (page 6, line 116).*

14. Fig. 2. The authors should explain the two colors of ACP in the β -methylation reaction inset (i.e. that the reaction sequence occurs in two different modules of Art11). In addition, for the non-expert reader, the legend should include definitions of all domain name abbreviations (ACP, KS, KSo, etc.). The authors should also explicitly state what the colored carbons refer to (i.e. points at which the methyl groups have been removed by inactivation of the corresponding

cMT domains)

Response: *The reaction sequence occurred on differently colored ACP-bound intermediates in module 3 and module 4 were explained in the legend of Fig. 2. The carbons labeled with varied colors represent the locations where the methyl groups were absent by inactivation of the corresponding MT domains, which was added in the legend of Fig. 2. All domain abbreviations in Fig.2 were explained in the legend as shown below.*

'Figure 2. The proposed biosynthetic pathway of ARTs based on the two-polyketide-chain assembly model. The inset depicts the process that methyl groups are appended to C-5 and C-7 of the ACP-bound intermediates by β -branching system. The methyl group at C-5 is first installed on the β -carbonyl of nascent polyketide chain tethered on ACP (dark blue) in module 3. subsequently, C-7 methylation was formed on the intermediate tethered on module 4 ACP (orange) in a same manner. The absence of methyl groups at C-2, C-14, and C-18 were highlighted with points in green, blue, and purple corresponding to the inactivated domains Art11MT, Art13MT, and Art14MT, respectively. ACP, acyl carrier protein; KS, ketosynthase; KS⁰, non-elongating ketosynthase; KR, ketoreductase; DH, dehydratase; ER, enoylreductase; MT, methyltransferase; HMGS, hydroxymethylglutaryl synthase.'

15. Page 7, line 122. It would be better to write that the feeding experiments 'suggested a possible origin for the free terminal carboxylic acid group' as being via oxidative cleavage, as the authors go on to show that it is generated by an alternative route (i.e. 'indicated' is misleading)

Response: *Thanks for the reviewer's suggestion! The sentence was rephrased as 'suggesting a possible origin for the free terminal carboxylic acid group by oxidative C-C bond cleavage of the polyketide chain and subsequent hydrolysis' (page 7, line 130).*

16. Page 7, line 125 (also Page 7, lines 135 and 139 and Page 8, line 145). The polyketide chain is not strictly-speaking 'split', but the BVMOs rather insert oxygen into the chain.

Response: *We agreed with the reviewer. The word 'split' was removed and the description that 'OocK and VdtE inserted an oxygen into the polyketide chain between the two carbons of one acetate unit' was used to make it more appropriate (page 7, line 133). Similar descriptions were also used in following text in the revised manuscript (page 8, lines 143, 147, and 152).*

17. Page 7, line 130. Again, appropriate reference needs to be made to the Supplementary materials.

Response: *The relevant Supplementary Figure 8 and Table 6 to ART 9B were added in the revised manuscript (page 7, line 139).*

18. Page 7, lines 120–134. Suggest using a synonym to 'expose' to avoid repetition, for example, 'reveal' or 'liberate'

Response: *The word 'exposing' was changed to 'liberating' in the revised manuscript as suggested.*

19. Page 8, line 142. But saying that this particular experiment was done 'carefully' , this would imply that the other experiments to which they refer (and by other researchers) were not.

Response: *Thank you for the suggestion! The word 'carefully' was removed to avoid unnecessary misunderstanding.*

20. Page 8, line 145. Suggest ‘established’ rather than ‘shaped’

Response: *Changed to ‘established’ as suggested.*

21. Page 9, line 171. Replace ‘resulted in’ with ‘yielded’

Response: *Changed as suggested.*

22. Page 9, line 175. By ‘typical type I PKS gene cluster’ do they mean ‘cis-AT PKS’? Again, this illustrates why more care needs to be taken to introduce the two large classes of modular PKSs from the beginning of the ms. In terms of the hybrid PKS-NRPS systems, do these incorporate cis-AT or trans-AT PKS components?

Response: *The introduction of cis-AT and trans-AT PKSs was added to the first paragraph of the introduction part.*

‘Specifically, a large family of modular PKSs is trans-AT (or AT-less) PKS, which does not have an AT domain in each extension module as cis-AT PKS but shares a standalone AT by multiple extension modules⁶⁻⁸.’ (page 3, line 40)

In terms of the hybrid PKS/NRPS gene clusters, they also incorporate cis-AT and trans-AT components. (Chen, H et al. Crit. Rev. Microbiol. 45, 162-181(2019); DOI: 10.1080/1040841X.2018.1514365) and (Piel, J. et al. Nat. Prod. Rep. 33, 231-316 (2016); DOI: 10.1039/c5np00125k) Therefore, we rewrote the sentence as ‘Among the 13 BGCs, 9 are trans-AT PKS gene clusters, 2 are hybrid cis-AT PKS/NRPS (non-ribosomal peptide synthetase) gene clusters, and the other 2 are hybrid trans-AT PKS/NRPS gene clusters.’ (page 9, line 181). Types of the different gene clusters in Fig. 4 were also revised appropriately according to the above description.

23. Page 10, line 192. Suggest, ‘The obtained data clearly show that ARTs only inhibit growth of Gram-positive...’

Response: *The sentence was changed as suggested in the revised manuscript (page 10, line 198).*

24. Figure 5. In this figure (and in the accompanying text, page 12), the authors have mixed chain initiation strategies for both cis-AT and trans-AT PKSs. In light of the fact that the two systems apparently have distinct evolutionary origins which likely contribute to the observed diversity, it would be appropriate to separate the strategies by PKS family. Again, definitions of domain/stand-alone enzyme acronyms should be provided. Also, ‘loading starter unit onto an ACP’; ‘by on-line modification of the starter unit’. Capitalize Transamidation, but not Type I. Also suggest as a title, ‘Selected initiation strategies...’, as this is not a complete list. Finally, citations to the relevant work demonstrating these strategies should be added.

Response: *Thanks for the reviewer’s suggestion! In the revised manuscript, we selected several representative examples of polyketide initiation strategies involving online modification of the starter unit and separated into two parts in Figure 5. Part a is on-line decarboxylation examples for cis-AT modular PKSs, and part b are on-line decarboxylation, transamidation, and methyl esterification examples for trans-AT modular PKSs. Accordingly, the title of Figure 5 was changed to ‘Selected initiation strategies involving on-line modification of the starter unit for cis-AT (a) and trans-AT (b) modular PKSs’. Definitions of domain/stand-alone enzyme acronyms were provided as suggested in figure legend. And, appropriate references were cited.*

The discussion part about polyketide initiation strategies was also revised as shown below (page 11, line 226). ‘Polyketide chain extension generally starts immediately after the starter unit is loaded; the initiation process includes preparation of the acyl-CoA starter unit (if it is not available from primary metabolism) and transacylation of the starter unit to the loading ACP^{10,12,32}. However, in some cases the starter unit must be modified after being loaded onto

an ACP in order to trigger polyketide chain extension. One classic example is on-line decarboxylation of the malonyl or methylmalonyl starter unit. In *cis*-AT modular PKSs, it is catalyzed by the N-terminal KS^Q domain of PKSs³³⁻³⁶(Fig. 5a), while in *trans*-AT modular PKSs, it is performed by a GCN5-related N-acetyltransferase (GNAT)-like domain^{37,38}(Fig. 5b). In addition, an on-line transamidation strategy has been proposed to initiate biosynthesis of glutarimide-containing compounds such as isomigrastatin and cycloheximide, although this mechanism requires further validation by *in vitro* studies^{39,40}(Fig. 5b).'

25. Page 10, line 199. Suggest, ‘...build on this work to establish a common...’

Response: *Changed as suggested.*

26. Page 10, line 201. Having introduced the concept of trans-AT PKSs earlier, it is not necessary to use the terms here (and indeed, the addition of a second one (‘AT-less’) will be confusing at this stage)

Response: *To avoid the confusion, ‘AT-less’ was showed only once when the term ‘trans-AT PKS’ was introduced for the first time in the revised manuscript (page 3, line 40).*

27. Page 13, line 258. Suggest, ‘However, such systems may yet be discovered by genome mining, while in the meantime, the results reported here should inspire synthetic biology efforts to generate novel polyketides incorporating terminal carboxylate groups.’ (And here, a reference to a review on PKS genetic engineering would be appropriate)

Response: *We rewrote this sentence as suggested and two reviews on PKS genetic engineering were referenced as ref. 50 and 51 in the revised manuscript.*

50. Hertweck, C. Decoding and reprogramming complex polyketide assembly lines: prospects for synthetic biology. *Trends Biochem. Sci.* **40**, 189-199 (2015). DOI: 10.1016/j.tibs.2015.02.001

51. Weissman, K. J. Genetic engineering of modular PKSs: from combinatorial biosynthesis to synthetic biology. *Nat. Prod. Rep.* **33**, 203-230 (2016). DOI: 10.1039/C5NP00109A

28. Page 20, line 460. There is an odd mixture here of cloning methodology, and protein expression/purification. These should be separated, and reference made to the appropriate table in the Supplementary.

Response: *The ‘construction of expression plasmids’ part and the ‘protein expression and purification’ part were described separately as suggested (page 19, line 386). And the appropriate Supplementary table was referred in the revised manuscript.*

29. Supplementary Table 1. The listed domain composition of Art11 isn’t consistent with Figure 2, as according to the figure, module 4 contains a tandem of ACP domains. Also, the authors could distinguish between the ECH1 and ECH2 homologs (Art20 and Art21).

Response: *Thanks very much! The domain components of Art11 were corrected in the revised Supplementary Table 1. Indeed, Art20 and Art21 are the homologs of ECH1 and ECH2 respectively based on the sequence similarities with characterized ECHs. They were distinguished in the revised Supplementary Table 1.*

30. Supplementary Fig. 6. Can the authors explain why the difference between the observed and calculated masses is larger in part a than in part b, for the same species?

Response: *In the original version, we just picked two peaks from the HR-MS data that are less than 5 parts per million*

(ppm) to the calculated mass of ART B, which is generally accepted by most journals. Thanks for the reviewer! After we scrutinized the mass data, the observed mass much closer to the calculated mass of ART B could be found in the HR-MS data of *B. subtilis* Δ art28/Bc-bioC. Thus, a more accurate mass data ($[M+H]^+=781.4159$) was used in part a of Supplementary Fig. 12 in the revised manuscript (the original Supplementary Fig.6).

31. Supplementary Fig. 7. Labeling should be added to this figure to identify to which of the multiple species potentially present in each assay the observed HPLC-MS peaks correspond. The legend to a is also incorrect, as in fact, malonyl-CoA methyl ester wasn't detected, but its expected mass is indicated. Same comment for part b of the legend.

Response: We are sorry for the confusion in Supplementary Fig. 13 (the original Supplementary Fig. 7)! In this figure, the Art28 catalyzed methylation of malonyl-CoA was tested. The results showed that Art28 can not use SAM as a methyl donor to form malonyl-CoA methyl ester. In part a, the left LC-MS data was the EIC for malonyl-CoA, which showed that, compared with the control assay with boiled Art28, the substrate malonyl-CoA was not clearly consumed in the malonyl-CoA+SAM+Art28 assay. The right LC-MS data was the EIC for malonyl-CoA methyl ester, which showed that no malonyl-CoA methyl ester was generated whether in the malonyl-CoA+SAM+Art28 assay or in the control assay. In part b, the left LC-MS data was the EIC for SAM, which showed that, compared with the control assay with boiled Art28, the substrate SAM was not clearly consumed in the malonyl-CoA+SAM+Art28 assay. The right LC-MS data was the EIC for SAH, which showed that no SAH was generated whether in the malonyl-CoA+SAM+Art28 assay or in the control assay.

In the revised manuscript, we labeled the peaks of malonyl-CoA and SAM as suggested. In addition, EIC for malonyl-CoA, EIC for malonyl-CoA methyl ester, EIC for SAM, and EIC for SAH were added to the related LC-MS data panel, respectively.

Typos/grammatical errors

1. Page 4, line 61: 'are reminiscent'

Response: It was corrected as suggested.

2. Page 5, line 85 (and Page 8, line 161). S-adenosyl-methionine (adenosyl not capitalized)

Response: They were corrected as suggested.

3. Page 6, line 116. 'Overall, the incorporation of all seven...'

Response: It was corrected as suggested.

4. Page 7, line 124. 'monooxygenases' (plural)

Response: It was corrected as suggested.

5. Page 7, line 135. Oocydin not Occydin

Response: It was corrected as suggested.

6. Page 8, line 140. , not ;

Response: It was corrected as suggested.

7. Page 9, line 181. ‘from the other MTs’

Response: *It was corrected as suggested.*

8. Page 10, line 185. ‘All five proteins...’

Response: *It was corrected as suggested.*

9. Page 10, line 205. ‘approximately’ instead of ‘about’

Response: *It was corrected as suggested.*

10. Page 10, line 205. ‘and Art14KS4 contains an Asn in place of the first His...’

Response: *It was corrected as suggested.*

11. Page 11, line 207. ‘have been observed’

Response: *It was corrected as suggested.*

12. Page 11, line 207. ‘We therefore propose that...’

Response: *It was corrected as suggested.*

13. Page 11, line 215. Remove ‘is’ before ‘originates’

Response: *It was corrected as suggested.*

14. Page 12, line 229. ‘such as isomigrastatin and cycloheximide, although this mechanism requires further validation by in vitro studies.’

Response: *It was corrected as suggested.*

15. Page 12, line 232. ‘and in the case of the ARTs,’

Response: *It was corrected as suggested.*

16. Page 12, line 237. ‘Art28-Art9-type’ (as the exploration is not limited to this specific pair)

Response: *It was changed as suggested.*

17. Page 12, line 240. ‘inhibitory activity’

Response: *It was corrected as suggested.*

(A few comments on the Materials and Methods, but the whole section needs careful proof-reading)

Response: *Thanks for the reviewer’s careful proof-reading, the whole section of the Material and Methods were carefully checked and revised.*

18. Page 15, line 350. ‘are listed’

Response: *It was corrected as suggested.*

19. Page 15, line 256. Remove ‘a’ before 1.1 kb

Response: *It was corrected as suggested.*

20. Page 15, line 365. Appropriate reference should be made to a table in the Supplementary for the primer sequences (and the same comment for the subsequent sections).

Response: *Supplementary Table 10 were referenced for primer sequences in relevant sections.*

21. Page 18, line 404. ‘downstream from the His735’

Response: *It was corrected as suggested.*

22. Supplementary Fig. 1 legend, line 101. ‘are highlighted’

Response: *It was corrected as suggested in S. Fig.5 legend (the original S. Fig. 1).*

Reviewer #2 (Remarks to the Author):

Li et al present their most recent findings on the biosynthesis of aurantinin and its congeners. The compounds and their biogenesis has been under investigation since the late 1970s and its biosynthesis has been (vaguely) elucidated before (citation 12, <https://doi.org/10.1021/acs.jafc.6b04455>) using genome analysis.

The current ms. tackles the assignment of the different (cryptic) ORF to specific functions in the biosynthesis and elucidate some interesting features. Apart from the possible (but not yet proven!) convergent biosynthetic layout (as also found in a number of other polyketides), the authors report one to the best of my knowledge new function: The loading unit malonic acid is selectively methylated by an methyltransferase which is essential for the activity of the whole PKS assembly line. Its significance for the self-protection of *B. subtilis* fmb609 is discussed.

The authors study this methyltransferase in vivo and to some extent in vitro and identify some homologs in other PKS gene clusters.

Indeed, the biosynthetic mechanism is interesting and worth publication. However, for publication in a high-impact journal more definitive answers to some pressing questions would be necessary. The authors themselves suggest some further experiments that would bolster up this study in the discussion. Thus, this article largely rests on the on-line methyl esterification initiation strategy, which indeed is unprecedented but at the same time rather a curiosity than a mechanism of great relevance for natural product research.

The more general message of this ms. thus is that researchers should look at least twice at unusual ORF – in vivo and in vitro.

Reviewer #3 (Remarks to the Author):

The manuscript by Li et al. describes a partial dissection of the Aurantinin (ARTs), which are antibacterial natural products generated by a trans-AT modular polyketide synthase system. The authors conducted a series of gene disruption (and allied complementation) experiments to determine the boundaries of the art biosynthetic gene cluster (BGC) that is comprised of 28 genes across ~80 kb. They also performed precursor incorporation studies to identify the subunits and source of methyl groups within the pathway. The aurantinin are structurally unique metabolites that contain a 6/7/8/5 fused ring system. Two hypotheses currently drive the studies to test whether a single linear polyketide chain is generated that subsequently forms the unique tetracyclic ring system, or two linear chains converge to create the core molecule. Figure 1 provides the data relating to the outer boundaries of the BGC, and a series of

comprehensive gene deletion and complementation experiments were conducted to support their characterization. A series of new molecules that represent intermediates in the pathway from gene deletion studies were isolated and characterized by MS and various rigorous NMR methods. The authors made the surprising finding that the starter unit of the pathway is in fact malonyl-CoA methyl ester, which they describe as a “protecting group” that renders the molecule inactive until that late stage of biosynthesis where the methyl group is removed by an esterase to unveil the carboxylate. It is this form of the molecule (ART B) that displays antibiotic activity against Gram positive bacteria. Biochemical studies were conducted on the recombinantly expressed methyltransferase (ART28) to demonstrate directly the formation of malonyl-CoA methyl ester. Direct studies were also performed on the corresponding esterase (ART9) to generate ART B (maximally active) from ART 9B (inactive).

There is significant merit in this study. The finding of a new type of starter unit that appears to “protect” the organism from its own antibiotic (self-resistance?) is important. Using in silico mining of natural product BGCs, the authors were able to identify numerous systems that appear to employ a similar biosynthetic strategy. This aspect of the work is perhaps the single most important contribution. Other aspects of the study (BGC sequencing, gene deletion, complementation) are relatively standard methods, but the authors were able to dissect some important aspects of the pathway.

Major Criticisms:

On the other hand, there are numerous aspects of the work that fall short. First, the most significant question is arguably the mode of assembly of the unusual 6/7/8/5 tetracyclic ring system. Although the authors present this as a challenge, the work leaves this question untouched.

Response: *We fully agree with the reviewer that one of the most intriguing structural features of ARTs is the complicated 6/7/8/5 tetracyclic system. In addition, the carboxyl terminus of ARTs is also very interesting, which was not frequently observed in polyketides. Actually, we started the biosynthesis study of ARTs aiming to solve both the formation mechanisms of the tetracyclic system and the carboxyl terminus.*

In this manuscript, we reported our findings about that ARTs recruit an unprecedented on-line methyl esterification strategy to initiation polyketide synthesis and expose the carboxyl terminus by a hydrolase at the very late stage of biosynthesis. The works described here is mostly about unraveling the ART polyketide chain assembly mechanism, which is not only meaningful to obtain more polyketides with a carboxylic end by genome mining and synthetic biology, but also is valuable foundation to further explore the formation mechanism of ART's tetracyclic system. Now, we are working on the 6/7/8/5 tetracyclic skeleton formation mechanism and have some progresses. Several potentially informative intermediates have been isolating from the art gene mutant strains. Expression and biochemical assays of relevant recombinant proteins were undergoing as well. Hopefully, we can share the tetracyclic system story with the community in the near future.

Second, although the hypothesis is presented that the malonyl-CoA methyl ester starter provides some form of self-protection to the producing microbe, the authors fail to address how this “prodrug” strategy operates at the molecule level on the proposed cellular target (see reference 12), or whether other types of cell targets may be involved in the antibacterial activity. In this respect, the authors' scholarship on this topic is inadequate. What other biosynthetic systems employ transient chemical modification as a mode of self-protection? Indeed, there are numerous examples in the literature and the authors need to cover this topic in the discussion section to enhance scholarship and to place their work in a proper perspective.

Response: *Thank you very much for the comments! We rewrote the third paragraph of the Discussion part and added*

some contents about the cellular target of ARTs, the possible mechanism of self-protection of the ARTs producer, and a summary of the natural products also adopting the pro-drug strategy. It is attached as below and the added sentences were underlined.

The esterification status of the carboxyl termini is critical for biological activity: the methyl ester ART 9B exhibits no inhibitory activity, but the free acid analog ART B can inhibit Gram-positive bacteria by disrupting cell membrane and causing leakage of intracellular components¹⁶. Introduction of the methyl ester group at the beginning of ART biosynthesis and hydrolyzing it at the end is reminiscent of traditional chemical synthetic protecting group strategies, and may similarly serve to prevent the unusual carboxylic acid from engaging in unwanted side reactions during the biosynthesis. In addition, the methyl ester group may also protect the producing organisms (Bacillus is also Gram-positive bacterium) from toxic ART intermediates. Methyl esterification of the ART intermediates may influence the interaction to cell membrane by changing their hydrophilia statuses dramatically. Such phenomenon was described previously for the ionophore antibiotic zincophorin, which also has a carboxyl terminus. The chemically synthesized zincophorin analogue with a terminal methyl ester modification exhibited no antibacterial activity⁴¹. Once the ART 'pro-drugs' are activated, they may be pumped out immediately by the efflux proteins, Art3 and Art27. This assumption is supported by the facts that (i) Art9 is a membrane located esterase with a transmembrane region (Supplementary Fig. 17); and (ii) ARTs are almost exclusively distributed in the fermentation broth but not inside cells (Supplementary Fig. 18). Similar protecting group strategy has been observed during the biosynthesis of a number of natural products and a versatile range of activation strategies have been recruited. For example, the 'pro-drug' of antibiotic xenocoumacin is activated via cleavage of an acylated D-asparagine by a membrane-anchored peptidase XcmG upon secretion⁴². In addition, zeamines⁴³, colibactin⁴⁴, and zwittermycin⁴⁵ are also activated by post-assembly proteolytic processing; while calyculin⁴⁶, oleandomycin⁴⁷, and naphthyridinomycin⁴⁸ are activated by dephosphorylation, deglycosylation, and oxidative processes, respectively.'

Minor Criticisms: The field has settled on the term "trans-AT" PKS as opposed to "AT-less" .

Response: As suggested, 'AT-less' was showed only once when the term 'trans-AT PKS' was introduced for the first time in the revised manuscript (page 3, line 40).

On line 36-37 the authors make the unsupported statement that the most common PKS starter unit is malonyl-CoA. How do they know this?

Response: Thanks for pointing out this improper statement! It was changed to 'malonyl-CoA is a frequently used starter unit' in the revised manuscript (page 3, line 45).

Reviewers' Comments:

Reviewer #1:

Remarks to the Author:

The authors have satisfactorily responded to the majority of my original comments.

I would note, however, the two remaining issues:

1. The 'co-linearity rule' that the authors newly reference in the introduction only applies to the cis-AT PKSs, for which there is a strong correlation between the composition of domains in the modules and the structures of the resulting products. In the case of the trans-AT PKSs, the much higher degree of domain diversity (including a large number of inactive domains), renders this relationship much less clear, and instead it is the phylogeny/substrate specificity of the KS domains downstream from each module which are predictive of product structure (indeed, this is the basis for the TransATor prediction pipeline which the authors used (doi:10.1038/nbt1379 and doi:0.1038/s41589-019-0313-7). This distinction should be clarified in the text.

2. Concerning the accurate mass of ART B (response 30 and Supplementary Figure 12), it doesn't seem reasonable to cherry pick the data to find an accurate mass that fits that which is calculated (why the mass of the same compound should differ between two strains isn't clear). Rather, for accurate mass determination, it is standard practice to take an average across the peak corresponding to the compound and to subtract appropriate background signal (signal to the left and right of the peak). The structure is not in question as NMR data for the compound are available (Supplementary Fig. 11), but any significant discrepancy revealed by this method between the experimental and calculated masses merits closer attention.

Please find below my thoughts on the authors' responses to reviewer 3's comments:

–The authors have essentially argued in their first response that while they didn't address all intriguing aspects of their biosynthesis, the demonstration of a novel antibiotic self-protection mechanism which appears to be general to a number of different pathways, nonetheless merits publication. Indeed, this was my overriding sentiment, as well as that of reviewer 2.

–It was previously known that the mature aurantinins act via membrane disruption. Here, the authors have provided convincing data to show that the 'prodrug' form is no more active than the DMSO control – and thus the methylation does confer self-resistance. They have nonetheless not addressed whether there are ADDITIONAL targets for the aurantinins (in either form), which is arguably outside of the scope of the ms.

(Note: in the revised Discussion, 'hydrophilia' is not the correct term)

–The third concern (which was also raised by me) has been adequately addressed as well in the revised version, by addition of a brief survey of previously-published instances of protecting group chemistries deployed in polyketide biosynthesis.

–I agree with Reviewer 3 that there is no need to mention the term 'AT-less', as it simply muddies the waters for those not expert in the field.

–The final comment has been adequately addressed.

Thanks for the reviewers' time and efforts!

In this revised version, we seek here to address the reviewers' concerns to the best of our ability. Reviewer suggestions warranting consideration are underlined and our specific responses are italicized. All changes were highlighted with yellow in the revised manuscript.

Reviewer 1's remarks:

The authors have satisfactorily responded to the majority of my original comments.

I would note, however, the two remaining issues:

1. The 'co-linearity rule' that the authors newly reference in the introduction only applies to the cis-AT PKSs, for which there is a strong correlation between the composition of domains in the modules and the structures of the resulting products. In the case of the trans-AT PKSs, the much higher degree of domain diversity (including a large number of inactive domains), renders this relationship much less clear, and instead it is the phylogeny/substrate specificity of the KS domains downstream from each module which are predictive of product structure (indeed, this is the basis for the TransATor prediction pipeline which the authors used (doi:10.1038/nbt1379 and doi:0.1038/s41589-019-0313-7). This distinction should be clarified in the text.

Response: *Thanks for the reviewer's excellent suggestion! A sentence was added to describe the distinction of trans-AT PKSs (page 3, line 42):*

'Notably, trans-AT PKS frequently deviates from this rule owing to the presence of modules with aberrant domain architecture or no apparent function. The phylogeny and substrate specificity of their KS domains can be applied for the prediction of product structures of trans-AT PKS^{10,11}.

In addition, the relevant references were added and reordered as ref. 10 and 11 in the revised manuscript.

*10. Helfrich, E. J. N. et al. Automated structure prediction of trans-acyltransferase polyketide synthase products. Nat. Chem. Biol. **15**, 813-821 (2019).*

*11. Nguyen, T. et al. Exploiting the mosaic structure of trans-acyltransferase polyketide synthases for natural product discovery and pathway dissection. Nat. Biotechnol. **26**, 225-233 (2008).*

2. Concerning the accurate mass of ART B (response 30 and Supplementary Figure 12), it doesn't seem reasonable to cherry pick the data to find an accurate mass that fits that which is calculated (why the mass of the same compound should differ between two strains isn't clear). Rather, for accurate mass determination, it is standard practice to take an average across the peak corresponding to the compound and to subtract appropriate background signal (signal to the left and right of the peak). The structure is not in question as NMR data for the compound are available (Supplementary Fig. 11), but any significant discrepancy revealed by this method between the experimental and calculated masses merits closer attention.

Response: *As suggested, we performed the recommended standard practice by taking an average and subtracting the background signals. The generated results are shown as below, which have less mass difference. Accordingly, it is used to replace the Supplement Figure 12 in the revised manuscript.*

Please find below my thoughts on the authors' responses to reviewer 3's comments:

–The authors have essentially argued in their first response that while they didn't address all intriguing aspects of their biosynthesis, the demonstration of a novel antibiotic self-protection mechanism which appears to be general to a number of different pathways, nonetheless merits publication. Indeed, this was my overriding sentiment, as well as that of reviewer 2.

Response: *Thanks for the reviewer's understanding! Hopefully, we can share the other stories of aurantinin with the community in the near future.*

–It was previously known that the mature aurantinins act via membrane disruption. Here, the authors have provided convincing data to show that the 'prodrug' form is no more active than the DMSO control – and thus the methylation does confer self-resistance. They have nonetheless not addressed whether there are ADDITIONAL targets for the aurantinins (in either form), which is arguably outside of the scope of the ms.

(Note: in the revised Discussion, 'hydrophilia' is not the correct term)

Response: *Thanks! We corrected the term 'hydrophilia' to 'hydrophilicity'. (page 13, line 258)*

–The third concern (which was also raised by me) has been adequately addressed as well in the revised version, by addition of a brief survey of previously-published instances of protecting group chemistries deployed in polyketide biosynthesis.

Response: *Thanks!*

–I agree with Reviewer 3 that there is no need to mention the term 'AT-less', as it simply muddies the waters for those not expert in the field.

Response: *To avoid the unnecessary confusion, the term 'AT-less' was removed from the text in the revised manuscript.*

–The final comment has been adequately addressed.

Response: *Thanks!*

Reviewers' Comments:

Reviewer #1:

Remarks to the Author:

I am now happy for the ms to proceed to publication.